# Quantile-Free Uncertainty Quantification in Graph Neural Networks

Soyoung Park [1]    Hwanjun Song [2]    Sungsu Lim [1]

## Abstract

Uncertainty quantification (UQ) in graph neural networks (GNNs) is crucial in high-stakes domains but remains a significant challenge. In graph settings, message passing often relies on strong assumptions such as exchangeability, which are rarely satisfied in practice, and achieving reliable UQ typically requires costly resampling or post-hoc calibration. To address these issues, we introduce Quantile-free Prediction Interval GNN (QpiGNN), a framework that builds on quantile regression (QR) to enable GNN-based UQ by directly optimizing coverage and interval width without requiring quantile inputs or post-processing. QpiGNN employs a dual-head architecture that decouples prediction and uncertainty, and is trained with label-only supervision through a quantile-free joint loss. This design allows efficient training and yields robust prediction intervals, with theoretical guarantees of asymptotic coverage and near-optimal width under mild assumptions. Experiments on 19 synthetic and real-world benchmarks show QpiGNN achieves average 22% higher coverage and 50% narrower intervals than baselines, while ensuring efficiency and robustness to noise and structural shifts.

## 1. Introduction

Graph Neural Networks (GNNs) are increasingly applied to node regression, where the goal is to predict target values for individual entities in graph-structured data. Node-level predictions have shown strong potential in high-stakes domains such as healthcare (Li et al., 2020) and criminal justice (Zhou et al., 2024), where reliability is critical. However, most existing GNNs rely on deterministic architectures that produce point estimates without uncertainty. Their

message-passing mechanisms can further propagate and amplify data biases (Jiang et al., 2024; Lin et al., 2024), increasing risks in real-world decision-making (Kwon et al., 2022). Although uncertainty-aware modeling has gained attention, uncertainty quantification (UQ) in GNN regression remains underexplored, hindering trustworthy applications. This typically involves prediction intervals that capture plausible ranges (Huang et al., 2023; Pouplin et al., 2024).

Recent works on UQ in GNNs can be broadly categorized into Bayesian and frequentist approaches (Chen et al., 2024; Wang et al., 2024a). Bayesian approaches (Zhao et al., 2020; Stadler et al., 2021) estimate posterior distributions but suffer from scalability issues and sensitivity to prior specification. Frequentist approaches are generally more efficient, but they often rely on resampling or post-hoc calibration, which can be costly or unstable in graph. Resampling-based methods (Kang et al., 2022; Liao et al., 2023) incur computational overhead due to repeated inference, while post-hoc calibration methods (Huang et al., 2023) require calibration sets and additional processing. In particular, these approaches often rely on strong assumptions such as exchangeability, which are often violated in graphs due to structural dependencies (Zhou et al., 2020b).

These limitations motivate the development of alternative frequentist approaches for GNN regression that avoid strong assumptions and post-processing while remaining computationally practical. We consider quantile regression (QR) (Koenker & Bassett Jr, 1978) as a promising alternative in graphs, due to its ability to handle non-Gaussian and heteroscedastic targets without restrictive distributional assumptions. Unlike resampling- or post-hoc methods, QR directly estimates conditional quantiles, avoiding repeated inference and reducing additional calibration steps. As a result, it provides a robust alternative for UQ in GNNs, where relational dependencies often violate standard assumptions (Ma et al., 2022; Angelopoulos & Bates, 2023).

Standard QR assumes i.i.d. data and requires quantile-level inputs. Specifically, the model either takes a quantile parameter $\tau$ (e.g., $\tau = 0.05, 0.95$) as input or trains separate predictors for each quantile to estimate the conditional quantile function $f(\mathbf{X}; \tau)$. This design increases model complexity and introduces issues such as quantile crossing (Zhou et al., 2020a). When combined with graph message passing,

[1]Department of Computer Science and Engineering, Chungnam National University, Daejeon, Korea [2]Department of Industrial and Systems Engineering, KAIST, Daejeon, Korea. Correspondence to: Sungsu Lim <sungsu@cnu.ac.kr>.

*Proceedings of the 43ʳᵈ International Conference on Machine Learning*, Seoul, South Korea. PMLR 306, 2026. Copyright 2026 by the author(s).

quantile supervision entangles node representations and often yields poorly calibrated, non-compact intervals (Rusch et al., 2023). Moreover, QR-based interval estimation methods that have not been applied to GNNs still rely on explicit quantile-level inputs or fail to distinguish predictions from uncertainty (Tagasovska & Lopez-Paz, 2019; Pouplin et al., 2024), which limits their stability and expressiveness in graph settings.

Motivated by limitations of existing methods, we develop a frequentist framework for UQ in GNNs that avoids strong assumptions such as exchangeability. To this end, we propose **Quantile-free Prediction Interval GNN (QpiGNN)**, a framework for UQ in GNNs through prediction interval estimation. QpiGNN is built on two key ideas. First, a *dual-head architecture* that separates prediction from uncertainty estimation, reducing oversmoothing and entanglement. Second, a *quantile-free joint loss* directly optimizes coverage and interval width from label-only supervision, eliminating the need for quantile-level inputs and post-processing. Collectively, these components enable stable training and calibrated intervals while ensuring coverage and near-optimal width under assumptions in graphs. The our code is available at `https://github.com/sybeam27/QpiGNN`, and our main contributions are summarized as follows:

- **Quantile-free UQ for GNNs**: We analyze the structural limitations of applying QR-based interval estimation to GNNs and introduce QpiGNN, the first quantile-free framework that enables calibrated and compact node-level uncertainty quantification.

- **Framework Design and Theory**: We develop a dual-head architecture with a quantile-free joint loss that decouples prediction from uncertainty, mitigates oversmoothing, and—under mild assumptions—offers theoretical guarantees of coverage and near-optimal width.

- **Empirical Results**: Across 19 diverse synthetic and real-world benchmarks, QpiGNN achieves on average 22% higher coverage and 50% narrower intervals than competitive baselines, while still remaining efficient and robust to noise and structural shifts.

## 2. Related Work

### 2.1. Uncertainty Quantification in GNNs

UQ in GNNs has been studied through ensemble, Bayesian, and frequentist approaches. Ensemble methods (Bazhenov et al., 2022; Wang et al., 2024b) improve robustness via multiple predictions but are costly and sensitive to shifts, limiting scalability in large graphs. Bayesian methods (Zhao et al., 2020; Stadler et al., 2021) provide principled probabilistic estimates by modeling posterior uncertainty but often face scalability issues and sensitivity to prior specification. Frequentist methods (Kang et al., 2022; Liao et al.,

2023) are more efficient but often depend on resampling or post-hoc calibration, causing significant overhead. Among such post-hoc approaches, Conformal Prediction (CP) provides distribution-free coverage guarantees but relies on exchangeability (often simplified as i.i.d.), an assumption that can be violated in graph data (Vovk et al., 2005). Although recent works attempt to relax these via partial exchangeability (Huang et al., 2023; Zhao et al., 2024), CP remains sensitive to heterogeneity such as hubs and heterophily, highlighting the need for alternative approaches.

### 2.2. Quantile Regression

QR models asymmetric and heteroscedastic targets without strong distributional assumptions (Koenker & Bassett Jr, 1978). However, existing graph-based QR methods such as GSL-QR (Zhang et al., 2023) and PE-GQNN (de Amorim et al., 2024) primarily target prediction or representation learning rather than uncertainty estimation. Extensions of QR for interval estimation, such as SQR (Tagasovska & Lopez-Paz, 2019) and RQR (Pouplin et al., 2024), improve flexibility but are not designed for GNNs, often causing calibration failures or oversmoothing when combined with message passing. In contrast, we integrate QR into GNNs in a quantile-free manner, removing explicit quantile-level inputs and post-processing, and thereby achieving calibrated and robust interval prediction for graphs.

## 3. Preliminaries and Background

This section introduces node-wise uncertainty quantification in graph regression via prediction intervals, along with the notations and background on QR and its extensions.

### 3.1. Node-wise Prediction Intervals in GNNs

Let $G = (\mathcal{V}, \mathcal{E})$ be a graph, where $\mathcal{V}$ is the set of nodes with $n = |\mathcal{V}|$, and $\mathcal{E} \subseteq \mathcal{V} \times \mathcal{V}$ is the set of edges. Each node $v \in \mathcal{V}$ is associated with a feature vector $\mathbf{x}_v \in \mathbb{R}^d$, where $d$ is the dimensionality of node features. The feature matrix is denoted by $\mathbf{X} \in \mathbb{R}^{n \times d}$. Each node also has a scalar regression target $y_v \in \mathbb{R}$, and the full target vector is $\mathbf{y} = [y_1, y_2, \ldots, y_n]^\top \in \mathbb{R}^n$.

The goal is to predict, for each node $v$, a *prediction interval* $[\hat{y}_v^{\text{low}}, \hat{y}_v^{\text{up}}]$ such that $\hat{y}_v^{\text{low}} \leq y_v \leq \hat{y}_v^{\text{up}}$ holds with high probability. The prediction intervals should be tight, node-wise adaptive, and avoid conservative bounds that are uniformly applied across all nodes.

GNNs update node representations by aggregating information from their local neighborhoods. The representation of node $v$, denoted as $\mathbf{h}_v$, is used to estimate the target value $y_v$. At layer $l$, the representation is updated as follows:

$$\mathbf{h}_v^{(l)} = \sigma\left(W^{(l)} \cdot \text{AGG}\left(\{\mathbf{h}_u^{(l-1)} \mid u \in \mathcal{N}(v)\}\right)\right), \quad (1)$$

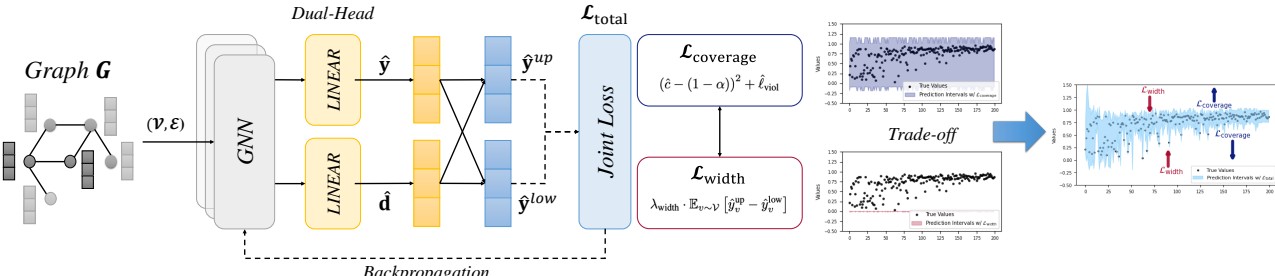

*Figure 1.* **Overview of Quantile-free Prediction Interval Graph Neural Network.** QpiGNN estimates node-wise prediction intervals using a *dual-head GNN* trained with a *quantile-free joint loss* that trades off coverage and compactness. One head predicts the target value $\hat{y}$, while the other predicts the interval width $\hat{d}$. The loss includes a coverage term $\mathcal{L}$coverage (⬆) encouraging wide enough intervals to maintain coverage, and a width term $\mathcal{L}$width (⬇) promoting tighter intervals. As illustrated, models trained with only the coverage term produce overly wide intervals, whereas the full joint loss yields tighter, locally adaptive intervals that still satisfy the target coverage $1 - \alpha$.

where $\mathrm{AGG} \in \{\mathrm{MEAN}, \mathrm{SUM}, \mathrm{MAX}\}$ is an aggregation function and $\sigma$ a non-linearity. Aggregation varies by architecture, using mean, sum, or attention (e.g., Graph-SAGE (Hamilton et al., 2017), GCN (Kipf & Welling, 2017)). Here, $\mathcal{N}(v)$ denotes the neighbors of node $v$.

### 3.2. Quantile Regression and Extensions

QR estimates the conditional quantile function $q_\tau(x)$ with $P(y \leq q_\tau(x)) = \tau$, where $\tau \in (0, 1)$ is the quantile level. Prediction intervals are obtained by setting the bounds to $\tau^{\mathrm{low}} = \alpha/2$ and $\tau^{\mathrm{up}} = 1 - \alpha/2$, where $\alpha$ is the miscoverage rate and $1 - \alpha$ the target coverage (e.g., $\alpha = 0.1$ yields a 90% interval). A limitation of QR is quantile crossing (Zhou et al., 2020a), where predictions at different quantile levels violate monotonicity, causing lower quantiles to exceed higher ones. To overcome this, extensions of QR have been proposed.

Tagasovska & Lopez-Paz (2019) proposed Simultaneous Quantile Regression (SQR), which jointly estimates multiple quantiles using a single model $f_\theta(x, \tau)$ conditioned on input and quantile level. Rather than training separate models for each quantile, SQR optimizes a objective. The standard QR loss, known as the pinball loss, is defined as:

$$\mathcal{L}_{\mathrm{QR}}(y, \hat{y}) = (\tau - \mathbb{I}(y < \hat{y}))(y - \hat{y}), \quad (2)$$

where $\tau \in (0, 1)$ is a quantile level. In contrast, SQR minimizes the expected pinball loss over a continuous range of quantiles $\tau \sim \mathcal{U}(0, 1)$:

$$\mathcal{L}_{\mathrm{SQR}} = \mathbb{E}_{\tau \sim \mathcal{U}(0,1)}[\mathcal{L}_{\mathrm{QR}}(y, f_\theta(x, \tau))]. \quad (3)$$

This formulation mitigates quantile crossing and improves parameter efficiency by sharing representations across quantile levels. However, our experiments show that SQR suffers from instability and calibration failure when combined with the message passing of GNNs as defined in Equation (1).

Beyond quantile-based approaches, recent extensions have explored predicting prediction intervals directly without requiring pre-defined quantile inputs $\tau$. Pouplin et al. (2024) proposed Relaxed Quantile Regression (RQR), developed for MLPs, which estimates prediction intervals by learning the conditional center and spread without relying on pre-defined quantile-level inputs. RQR predicts input-dependent bounds $[\hat{y}^{\mathrm{low}}(x), \hat{y}^{\mathrm{up}}(x)]$ and optimizes them for target coverage and width using a width-regularized objective:

$$
\begin{aligned}
\mathcal{L}_{\mathrm{RQR\text{-}W}} = {} & \big(\alpha + 2\lambda - \mathbb{I}\big[\hat{y}^{\mathrm{low}} \leq y \leq \hat{y}^{\mathrm{up}}\big]\big) \\
& \times (y - \hat{y}^{\mathrm{low}})(y - \hat{y}^{\mathrm{up}}) + \frac{\lambda}{2}\big(\hat{y}^{\mathrm{up}} - \hat{y}^{\mathrm{low}}\big)^2,
\end{aligned} \quad (4)
$$

where $\lambda \geq 0$ controls the trade-off between coverage and compactness and $\mathbb{I}(\cdot)$ is the indicator function. The first term enforces coverage, while the second penalizes width.

While effective in tabular MLPs, our experiments reveal that directly extending RQR to GNNs introduces significant challenges. Message passing tends to produce overly smooth and global intervals (Rusch et al., 2023), limiting node-wise adaptivity. Moreover, the single-head design of RQR entangles the learning of center and spread, reducing representation flexibility and degrading calibration performance under graph-structured data. To improve interval validity in graph-based tasks, we add an ordering penalty $\gamma_{\mathrm{order}}$, which facilitates comparison with our proposed model:

$$\mathcal{L}_{\mathrm{RQR}^{adj.}} = \mathcal{L}_{\mathrm{RQR\text{-}W}} + \gamma_{\mathrm{order}} \cdot \mathrm{ReLU}(\hat{\mathbf{y}}^{\mathrm{low}} - \hat{\mathbf{y}}^{\mathrm{up}}), \quad (5)$$

where $\gamma_{\mathrm{order}} \geq 0$ is a hyperparameter. This penalty, *absent in the original formulation*, is added as a practical fix: applying RQR to GNNs causes quantile crossing that hinders evaluation, rather than reflecting any graph-specific mechanism.

# 4. QpiGNN: Quantile-free Prediction Interval Graph Neural Network

## 4.1. Problem Formulation

We address uncertainty quantification in graph-structured data by learning calibrated prediction intervals without relying on quantile inputs, resampling, and post-processing. Given a graph $G$ with node features $\mathbf{X}$, the objective is to learn a function $f : \mathcal{V} \to \mathbb{R}^2$ that assigns each node $v$ a prediction interval $f(v) = [\hat{y}_v^{\text{low}}, \hat{y}_v^{\text{up}}]$, such that:

$$\mathbb{P}\left(\hat{y}_v^{\text{low}} \leq y_v \leq \hat{y}_v^{\text{up}}\right) \geq 1 - \alpha, \quad \mathbb{E}\left[\hat{y}_v^{\text{up}} - \hat{y}_v^{\text{low}}\right] \text{ minimized.}$$

Here, $1 - \alpha$ denotes the target coverage, the probability that true values fall within the predicted intervals. Thus, the learned intervals should be both *calibrated*—achieving the target coverage across nodes—and *compact*—minimizing their average width. We propose QpiGNN, a GNN-based framework for node-wise uncertainty quantification via calibrated prediction intervals.

## 4.2. Dual-head GNN for Calibrated Prediction Intervals

GNNs generate node representations by aggregating neighborhood information (Hamilton et al., 2017; Kipf & Welling, 2017). While effective, single-head architectures that jointly estimate prediction bounds can suffer from oversmoothing (Li et al., 2018), making it difficult to capture uncertainty. To mitigate this, QpiGNN employs a dual-head architecture that decouples the estimation of prediction $\hat{\mathbf{y}}$ and interval width $\hat{\mathbf{d}}$, allowing each to be optimized for a specific objective—accuracy for $\hat{\mathbf{y}}$, and coverage for $\hat{\mathbf{d}}$.

Separating the two heads also prevents representational conflicts and gradient interference between prediction and uncertainty estimation, enabling each branch to learn its own function class more effectively. Similar dual-head structures have demonstrated strong performance in heteroscedastic and Bayesian regression (Kendall & Gal, 2017; Lakshminarayanan et al., 2017). As shown in Figure 1, this design enables expressive, node-wise uncertainty modeling by structurally separating prediction and uncertainty components, with the full training algorithm provided in Appendix A.

The effectiveness of this design is validated through an ablation study in Section 5.2.7. Across nine synthetic datasets, the dual-head GNN with a learnable margin outperforms fixed-margin or single-output variants, achieving better empirical trade-offs by maintaining high coverage with significantly narrower intervals. QpiGNN uses a GNN encoder to compute node embeddings $\mathbf{H} = \text{GNN}(\mathbf{X}, \mathcal{E})$, followed by two linear heads $\hat{\mathbf{y}} = \mathbf{W}_{\text{pred}}\mathbf{H} + \mathbf{b}_{\text{pred}}$, $\hat{\mathbf{d}} = \text{Softplus}(\mathbf{W}_{\text{diff}}\mathbf{H} + \mathbf{b}_{\text{diff}})$. The softplus activation ensures $\hat{\mathbf{d}} > 0$, yielding half-widths, and the final prediction interval

for node $v$ is given by:

$$\hat{y}_v^{\text{low}} = \hat{y}_v - \hat{d}_v, \quad \hat{y}_v^{\text{up}} = \hat{y}_v + \hat{d}_v.$$

This architecture allows QpiGNN to estimate calibrated prediction intervals end-to-end—without requiring quantile-level input, resampling, or post-processing.

## 4.3. Joint Loss for Compact Prediction Intervals

To train QpiGNN to produce prediction intervals that are both calibrated and compact, we define a loss function $\mathcal{L}_{\text{total}}$ consisting of a coverage term $\mathcal{L}_{\text{coverage}}$ and a width regularization term $\mathcal{L}_{\text{width}}$.

$$\mathcal{L}_{\text{total}} = \underbrace{(\hat{c} - (1 - \alpha))^2 + \hat{\ell}_{\text{viol}}}_{\mathcal{L}_{\text{coverage}}} + \underbrace{\lambda_{\text{width}} \cdot \mathbb{E}_{v \sim \mathcal{V}}\left[\hat{y}_v^{\text{up}} - \hat{y}_v^{\text{low}}\right]}_{\mathcal{L}_{\text{width}}},$$

(6)

where the empirical coverage is defined as $\hat{c} := \mathbb{P}\left(\hat{y}_v^{\text{low}} \leq y_v \leq \hat{y}_v^{\text{up}}\right)$, and the empirical violation loss[1] $\hat{\ell}_{\text{viol}}$ is given by:

$$\mathbb{E}\left[|y_v - \hat{y}_v^{\text{low}}| \cdot \mathbb{I}[y_v < \hat{y}_v^{\text{low}}] + |y_v - \hat{y}_v^{\text{up}}| \cdot \mathbb{I}[y_v > \hat{y}_v^{\text{up}}]\right].$$

The violation penalty provides fine-grained feedback when predicted intervals fail to capture true targets.[2] The term $\lambda_{\text{width}}$ balances calibration against interval compactness.

Unlike RQR-W in Equation (4), which entangles coverage and width into a single conditional loss—often resulting in oversmoothed intervals in GNNs (Rusch et al., 2023)—our formulation separates these objectives. This separation yields more stable intervals under graph-based learning, as it directly optimizes empirical coverage and width over nodes. We further validate this in Appendix B by applying our loss to an RQR formulation, showing that our objective yields more stable intervals.

Specifically, it penalizes deviations from the target coverage $(1 - \alpha)$ via $(\hat{c} - (1 - \alpha))^2$, while independently regularizing interval width via $\lambda_{\text{width}} \cdot \mathbb{E}_{v \sim \mathcal{V}}[\hat{y}_v^{\text{up}} - \hat{y}_v^{\text{low}}]$. This disentangled design enables stable training and precise control over node-level uncertainty. Its linear additive penalty prevents excessive amplification and yields more stable behavior across datasets.[3] As shown in Figure 1, jointly minimizing $\mathcal{L}_{\text{coverage}}$ and $\mathcal{L}_{\text{width}}$ yields prediction intervals while maintaining the coverage.

---

[1]For notational compactness, an equivalent form is $\hat{\ell}_{\text{viol}} = \mathbb{E}_{v \sim \mathcal{V}}[\max(0, (y_v - \hat{y}_v^{\text{low}})(y_v - \hat{y}_v^{\text{up}}))]$; we retain the decomposed form for interpretability and slightly more stable gradients.

[2]In finite samples, the violation loss provides a useful training signal; once the model is calibrated, its effect diminishes and we omit it from the asymptotic analysis (Appendix C.4).

[3]Conversely, when the target range is narrow, multiplicative terms may operate more stably.

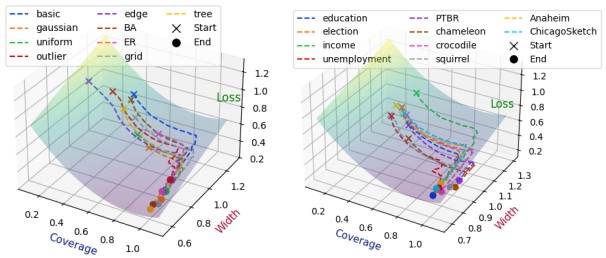

*Figure 2.* Loss convergence trajectories on synthetic and real-world datasets. The paths show progression from ✗ (start) to ● (end). Left: synthetic datasets. Right: real-world datasets.

### 4.3.1. ASYMPTOTIC AND FINITE-SAMPLE COVERAGE

A goal of QpiGNN is to ensure that predicted intervals achieve the coverage level $1 - \alpha$ at finite and asymptotic sample sizes. We provide justification for why the empirical coverage $\hat{c}$ remains close to the target under mild conditions.

**Proposition 4.1.** *Assume the following mild conditions: (i) the label noise $\varepsilon_v = y_v - f(x_v)$ is bounded and weakly dependent across nodes, (ii) the predicted mean $\hat{y}_v$ and interval half-width $\hat{d}_v$ converge in probability to their targets, and (iii) node embeddings remain sufficiently diverse. Then, as $N \to \infty$, the empirical coverage converges in probability to the target level $1 - \alpha$:*

$$\hat{c} \xrightarrow{P} 1 - \alpha, \quad \text{equivalently,}$$

$$\forall \varepsilon > 0, \ \lim_{N \to \infty} \mathbb{P}(|\hat{c} - (1 - \alpha)| > \varepsilon) = 0.$$

*Sketch of Proof.* Define each prediction interval as $[\hat{y}_v^{\text{low}}, \ \hat{y}_v^{\text{up}}]$ and let $Z_v := \mathbb{I}[\hat{y}_v^{\text{low}} \leq y_v \leq \hat{y}_v^{\text{up}}]$. Under assumptions (i)–(iii), the expected coverage satisfies $\mathbb{E}[Z_v] \to 1 - \alpha$. By the Weak Law of Large Numbers (WLLN) (Penrose & Yukich, 2003; Gama & Ribeiro, 2019), the empirical coverage $\hat{c} = \frac{1}{N} \sum_{v=1}^{N} Z_v \xrightarrow{P} 1 - \alpha$. □

### 4.3.2. FINITE-SAMPLE COVERAGE

For finite samples, deviation of empirical coverage from the target can be controlled using classical concentration inequalities. Under localized message passing, the bounded-difference condition holds approximately, so the inequalities of McDiarmid (1989) and Hoeffding (1994) still apply up to a graph-dependent constant. This yields $|\hat{c} - (1 - \alpha)| = \mathcal{O}(1/\sqrt{N})$, implying that for moderate $N$, empirical coverage $\hat{c}$ remains close to the target $1 - \alpha$ with high probability, consistent with the stability observed in our experiments.

This section focuses on finite-sample guarantees derived from concentration inequalities rather than asymptotic arguments. As detailed in Appendix C.5, perturbing a single node affects the coverage estimator by at most $(1/N + \delta_G)$,

providing the theoretical basis for the approximate bounded-difference behavior used in our analysis. Moreover, our finite-sample guarantees bound the deviation $|\hat{c} - \mathbb{E}[\hat{c}]|$ even in the absence of exact independence, since the estimator exhibits controlled sensitivity to single-node perturbations, which provides the bounded-difference needed for applying concentration arguments in practical graph regimes.

### 4.3.3. OPTIMAL WIDTH UNDER COVERAGE

To encourage compact prediction intervals, we introduce a width penalty term $\mathcal{L}_{\text{width}}$, weighted by a parameter $\lambda_{\text{width}} \geq 0$, which can be tuned via Bayesian optimization (Snoek et al., 2012) to balance calibration (0.2–0.5; Appendix E). We adopt an L1-based width penalty, avoiding the instability of L2 losses under outliers (Pearce et al., 2018; Tagasovska & Lopez-Paz, 2019) (See Appendix F).

Theoretically, if the conditional distribution $P(y \mid x_v)$ is symmetric around the predicted mean $\hat{y}_v$, the minimum-width interval satisfying the coverage constraint $\hat{c} \geq 1 - \alpha$ is $[\hat{y}_v - d_v^*, \ \hat{y}_v + d_v^*]$, where $d_v^* = F_v^{-1}(1 - \alpha/2)$ is the quantile of $|y_v - \hat{y}_v|$. Since the true distribution is unknown, QpiGNN instead minimizes the total loss $\mathcal{L}_{\text{total}} = \mathcal{L}_{\text{coverage}} + \lambda_{\text{width}} \cdot \mathcal{L}_{\text{width}}$, which can be interpreted as a Lagrangian relaxation (Franceschi et al., 2019) of the constrained optimization problem. When $\hat{c}$ approaches the target $1 - \alpha$, the model prioritizes width reduction, yielding near-optimal intervals, as detailed in Appendix C.2.

### 4.3.4. CONVERGENCE PROPERTIES OF THE JOINT LOSS

The joint loss $\mathcal{L}_{\text{total}}$ is non-convex, but each component is continuous and piecewise smooth, which allows the use of standard stochastic approximation techniques. Following results from stochastic non-convex optimization (Ghadimi & Lan, 2013), we show that training with a diminishing learning rate ensures convergence to a stationary point. Specifically, under the assumptions that (i) each component is continuous and smooth, (ii) gradients are bounded, and (iii) the step size decays appropriately, the training dynamics satisfy $\lim_{t \to \infty} \mathbb{E} \left[ \|\nabla_\theta \mathcal{L}_{\text{total}}(\theta_t)\| \right] = 0$, implying that the parameters converge to a point where the expected gradient norm vanishes (Kingma & Ba, 2015; Reddi et al., 2018).

This property is crucial given the hybrid nature of the loss, which balances interval width and coverage. As shown in Figure 2, training first reduces sharp coverage-violation penalties and then progressively narrows intervals, following a natural trajectory toward minimizing the overall loss while maintaining target coverage. A formal proof and empirical validation are provided in Appendix C.3.

**Limitations and Practical Considerations.** Our theoretical guarantees rely on standard but idealized assumptions such as node-wise independence and symmetric conditional

distributions. In real-world graphs, message passing induces weak dependencies, and target distributions may be skewed or heavy-tailed (Jin et al., 2020; Verma & Zhang, 2019). As noted in Appendix C.5, these weak dependencies do not significantly impair convergence of the coverage estimator $\hat{c}$, and the bounded-difference condition for McDiarmid's inequality holds approximately (McDiarmid, 1989). Moreover, QpiGNN achieves valid coverage and compact intervals under moderate distributional shifts, consistent with findings in robust quantile regression (Tagasovska & Lopez-Paz, 2019) and relaxed quantile modeling under asymmetric noise (Pouplin et al., 2024). Future work may extend our theory to graph-dependent mixing, adaptive tuning of $\lambda_{\text{width}}$ (e.g., via annealing or meta-learning), and asymmetric interval construction to better handle skewed target distributions. Furthermore, the sensitivity of the dual-head architecture to the choice of underlying GNN backbone warrants investigation. Despite simplifications, QpiGNN outperforms baselines in 7 of 10 real-world datasets (Sec. 5), indicating robust performance.

# 5. Experiments

## 5.1. Experimental Settings

### 5.1.1. DATASETS

We evaluate QpiGNN on 19 datasets. The top half of Table 1 presents synthetic graphs, while the bottom half lists real-world node-level regression benchmarks (Rozember-czki et al., 2021). Detailed descriptions of all datasets are provided in Appendix G.

### 5.1.2. BASELINES

We compare QpiGNN with six baselines: SQR-GNN, an extension of the MLP-based SQR to GNNs; RQR$^{adj.}$-GNN, which extends RQR to GNNs by enforcing interval ordering in Equation (5); and CF-GNN (Huang et al., 2023). For CF-GNN, we report both our reimplementation[4] and the original version with tuned hyperparameters[5], denoted CF-GNN$^{opt.}$. We also include Evidential Regression (ER)-GNN, a deterministic method that estimates uncertainty by predicting evidential parameters (Amini et al., 2020). For comparison with approximate Bayesian inference methods, we also evaluate BayesianNN (Kendall & Gal, 2017) and MC Dropout (Gal & Ghahramani, 2016).

### 5.1.3. METRICS

For evaluation, we primarily report Prediction Interval Coverage Probability (PICP) (Rana et al., 2015) and Mean Pre-diction Interval Width (MPIW) (Khosravi et al., 2010a), which together assess calibration and compactness. Additional metrics are included in Appendix H.

### 5.1.4. IMPLEMENTATION

All models target $1-\alpha = 0.90$, and use GraphSAGE with two layers and hidden size 64. We train with the Adam optimizer (learning rate and weight decay = $10^{-3}$) for 500 epochs, averaging over 5 or 10 runs. MC Dropout uses 0.2 dropout rate and 100 stochastic passes. RQR$^{adj.}$-GNN is trained with fixed $\lambda = 1.0$ [6] and order penalty $\gamma_{\text{order}} = 1.0$. All baselines are trained with standard MSE loss. Full details are in Appendix I and our codebase.

## 5.2. Experimental Results

### 5.2.1. QUANTITATIVE EVALUATION

Table 1 reports PICP and MPIW on 19 datasets under three width penalties: default ($\lambda^{0.5}$), conservative ($\lambda^{0.1}$), and Bayesian-optimized ($\lambda^{opt.}$). On synthetic graphs, $\lambda^{opt.}$ reliably meets target coverage with the lowest MPIW in most cases, while $\lambda^{0.5}$ yields tight but sometimes under-covering intervals and $\lambda^{0.1}$ attains near-perfect coverage at the cost of wider widths. On real graphs, $\lambda^{0.5}$ provides the narrowest intervals, whereas $\lambda^{opt.}$ balances coverage and compactness (e.g., *Chameleon*, *Squirrel*). Against baselines, QpiGNN shows superior trade-off: BayesianNN ensures coverage with wide intervals, MC Dropout is narrow yet unreliable, and SQR-GNN, RQR$^{adj.}$-GNN, and CF-GNN often suffer poor calibration or unstable widths. SQR-GNN can exhibit instability on heterophily graphs, but if slight under-coverage is permitted—as is sometimes allowed outside standard UQ conventions—QR-based baselines can also demonstrate competitive performance. QpiGNN with $\lambda^{opt.}$ improves coverage by 22% and reduces width by 50% on average, with further results in Appendix J.

### 5.2.2. QUALITATIVE EVALUATION

Fig. 3 illustrates prediction intervals of six models on two representative datasets: *Tree* (synthetic) and *Anaheim* (real-world). On the *Tree* dataset, SQR and MC Dropout produce tight but under-covering intervals, missing several true values. In contrast, QpiGNN slightly increases its interval width to meet the target coverage ($1-\alpha = 0.90$), demonstrating a balance between calibration and compactness. RQR yields globally uniform intervals, while BayesianNN and CF-GNN achieve full coverage at the cost of significantly wider intervals. Similar trends are observed on the *Anaheim* dataset, confirming the consistency of QpiGNN across data regimes. These visual comparisons highlight

---

[4]Same GNN architecture, layers, hidden size, and epochs.

[5]Including $\tau$, learning rate, target interval size, loss weights, and architecture.

[6]Using $\lambda = 0.5$ or 0.1 yielded prediction intervals performance differences within 0.01 compared to $\lambda = 1$.

*Table 1.* Prediction intervals performance (PICP/MPIW) on 19 synthetic and real datasets. Models are grouped by dataset and evaluated on test splits. **Bold** indicates coverage above the 90% target $(1-\alpha)$, and underline marks the lowest MPIW among those. This highlights models achieving both calibration and compactness. $\lambda^{opt.}$ denotes the width penalty selected via Bayesian optimization.

| Model | Basic | | Gaussian | | Uniform | | Outlier | | Edge | | BA | | ER | | Grid | | Tree | |
|---|---|---|---|---|---|---|---|---|---|---|---|---|---|---|---|---|---|---|
| | PCIP | MPIW | PCIP | MPIW | PCIP | MPIW | PCIP | MPIW | PCIP | MPIW | PCIP | MPIW | PCIP | MPIW | PCIP | MPIW | PCIP | MPIW |
| SQR | 0.85 | 0.33 | 0.88 | 0.50 | 0.88 | 0.51 | **0.90** | 0.10 | **0.91** | 0.32 | 0.81 | 0.72 | 0.75 | 0.60 | 0.78 | 0.53 | 0.80 | 0.26 |
| RQR$^{adj.}$ | **0.90** | 0.82 | 0.88 | 0.53 | **0.90** | 0.68 | **0.90** | 0.36 | **0.93** | 0.83 | 0.78 | 0.75 | 0.88 | 0.77 | 0.72 | 0.48 | 0.85 | 0.68 |
| BayesianNN | **1.00** | 3.01 | **1.00** | 2.98 | **1.00** | 3.00 | **1.00** | 2.95 | **1.00** | 3.06 | **1.00** | 3.08 | **1.00** | 3.01 | **1.00** | 3.01 | **1.00** | 3.00 |
| MC dropout | **0.99** | 0.32 | 0.55 | 0.20 | 0.65 | 0.26 | 0.58 | 0.06 | **1.00** | 0.30 | 0.67 | 0.26 | 0.76 | 0.23 | 0.33 | 0.16 | 0.64 | 0.20 |
| CF-GNN | **0.92** | 1.90 | **0.91** | 2.90 | **0.90** | 3.04 | **0.93** | 1.92 | **0.92** | 1.78 | **0.90** | 68.27 | **0.90** | 17.15 | **0.94** | 3.18 | **0.93** | 0.97 |
| QpiGNN ($\lambda^{0.5}$) | 0.89 | 0.30 | **0.92** | 0.55 | 0.88 | 0.43 | 0.89 | 0.47 | **0.94** | 0.39 | **0.98** | 0.49 | **0.98** | 0.63 | **0.98** | 0.87 | **0.96** | 0.39 |
| QpiGNN ($\lambda^{0.1}$) | **0.98** | 0.93 | **0.99** | 0.84 | **0.99** | 0.89 | **0.99** | 0.75 | **1.00** | 0.97 | **0.98** | 1.01 | **0.99** | 0.92 | **0.98** | 0.98 | **1.00** | 0.59 |
| QpiGNN ($\lambda^{opt.}$) | **0.90** | 0.30 | **0.95** | 0.64 | **0.93** | 0.62 | **0.90** | 0.49 | **0.94** | 0.54 | **0.98** | 0.48 | **0.98** | 0.63 | **0.99** | 0.93 | **0.96** | 0.39 |

| Model | Education | | Election | | Income | | Unemploy. | | Twitch | | Chameleon | | Crocodile | | Squirrel | | Anaheim | | Chicago | |
|---|---|---|---|---|---|---|---|---|---|---|---|---|---|---|---|---|---|---|---|---|
| | PCIP | MPIW | PCIP | MPIW | PCIP | MPIW | PCIP | MPIW | PCIP | MPIW | PCIP | MPIW | PCIP | MPIW | PCIP | MPIW | PCIP | MPIW | PCIP | MPIW |
| SQR | 0.88 | 0.32 | 0.89 | 0.47 | 0.86 | 0.22 | 0.87 | 0.33 | 0.30 | 0.03 | 0.37 | 0.01 | 0.44 | 0.01 | 0.22 | 0.01 | 0.88 | 0.32 | 0.87 | 0.21 |
| RQR$^{adj.}$ | 0.87 | 0.49 | 0.89 | 0.54 | 0.89 | 0.36 | **0.90** | 0.38 | **0.91** | 0.42 | 0.86 | 0.15 | 0.87 | 0.08 | 0.89 | 0.15 | 0.85 | 0.50 | 0.88 | 0.30 |
| BayesianNN | **1.00** | 2.96 | **1.00** | 2.98 | **1.00** | 2.97 | **1.00** | 2.98 | **1.00** | 3.07 | **1.00** | 2.95 | **1.00** | 3.07 | **1.00** | 2.97 | **1.00** | 2.94 | **1.00** | 2.99 |
| MC dropout | 0.40 | 0.11 | 0.48 | 0.18 | 0.45 | 0.09 | 0.41 | 0.09 | **0.91** | 0.15 | 0.47 | 0.02 | 0.46 | 0.01 | 0.31 | 0.02 | 0.50 | 0.11 | 0.34 | 0.07 |
| CF-GNN | 0.88 | 2.78 | 0.89 | 1.08 | **0.92** | 3.48 | **0.90** | 3.16 | **0.92** | 3.53 | - | - | - | - | - | - | **0.90** | 3.22 | **0.90** | 3.12 |
| CF-GNN$^{opt.}$ | **0.90** | 3.10 | **0.91** | 0.94 | **0.91** | 2.92 | 0.89 | 2.61 | 0.89 | 2.34 | - | - | - | - | - | - | **0.90** | 2.82 | **0.91** | 2.26 |
| QpiGNN ($\lambda^{0.5}$) | **0.99** | 0.57 | **0.98** | 0.77 | **0.99** | 0.41 | **1.00** | 0.74 | 0.59 | 0.08 | 0.51 | 0.03 | **0.92** | 0.08 | 0.73 | 0.07 | **0.92** | 0.39 | **0.97** | 0.36 |
| QpiGNN ($\lambda^{0.1}$) | **0.99** | 0.90 | **1.00** | 0.97 | **1.00** | 0.72 | **1.00** | 0.93 | **0.98** | 0.54 | **0.98** | 0.40 | **1.00** | 0.54 | **0.99** | 0.47 | **0.99** | 0.74 | **0.99** | 0.60 |
| QpiGNN ($\lambda^{opt.}$) | **0.99** | 0.59 | **0.98** | 0.77 | **0.99** | 0.44 | **1.00** | 0.73 | **0.94** | 0.36 | **0.96** | 0.23 | **0.97** | 0.16 | **0.96** | 0.18 | **0.93** | 0.40 | **0.98** | 0.36 |

(a) Tree synthetic dataset         (b) Anaheim real dataset

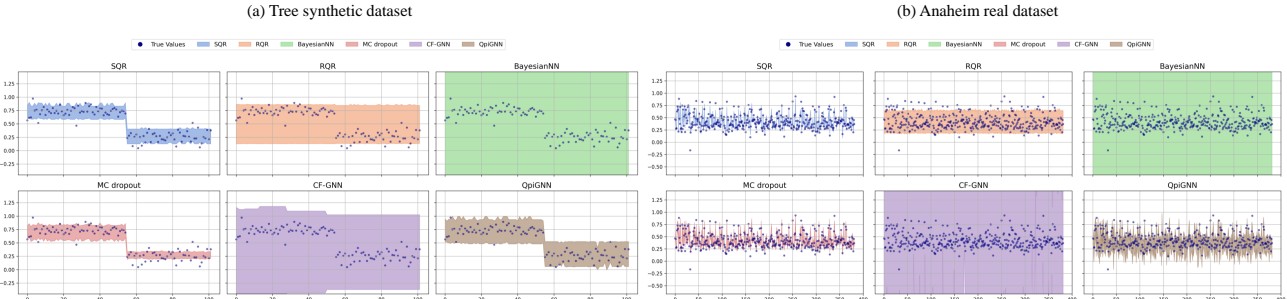

*Figure 3.* Prediction intervals comparison on the *Tree* synthetic dataset and the *Anaheim* real-world dataset. Models are trained with $\lambda_{\text{width}} = 0.5$ for 500 epochs.

QpiGNN's ability to adapt its intervals based on data uncertainty while maintaining strong calibration. Additional results for all 19 datasets are provided in Appendix K.

### 5.2.3. ROBUSTNESS TO PERTURBATIONS

We evaluate robustness under three types of perturbations on ER graphs in Table 2. Across all settings, QpiGNN consistently maintains valid coverage and compact intervals, outperforming SQR-GNN and RQR$^{adj.}$-GNN. Under feature noise, QpiGNN maintains high coverage (PICP $\approx 0.90$) with moderate widths, while SQR-GNN under-covers and RQR$^{adj.}$-GNN is less stable. For target noise, QpiGNN expands intervals to preserve near-perfect coverage (PICP $= 1.00$ at $\sigma = 0.3$, MPIW $\approx 1.05$), whereas SQR-GNN collapses early and RQR$^{adj.}$-GNN needs much wider intervals. With edge dropout, QpiGNN raises coverage ($0.84 \rightarrow 0.91$)

while keeping MPIW compact , outperforming SQR-GNN's poor coverage and RQR$^{adj.}$-GNN's wide intervals.

### 5.2.4. STRUCTURAL SHIFT GENERALIZATION

We evaluate robustness to structural shifts by training QpiGNN on one graph type and testing on others in Appendix M. This setting reflects realistic scenarios where graphs evolve, are partially observed, or reconstructed over time, making robustness to such changes critical for GNN-based UQ. As shown in Figure 4, models trained on expressive graphs like *BA* and *ER* generalize best, achieving PICP $\geq 0.89$ on most targets. In contrast, models trained on simpler sources (e.g., *Tree*, *Basic*) transfer poorly, especially to irregular graphs (e.g., PICP $< 0.2$ on *BA*). We also observe a coverage–width trade-off: *Basic* and *Tree* yield narrow but under-covering intervals, while *ER* strikes a better balance.

*Table 2.* Robustness analysis on ER graphs with feature/target noise and edge dropout. We compare QpiGNN, SQR-GNN, and RQR$^{adj.}$-GNN, reporting average PICP and MPIW over 5 runs.

| Perturbation Type | Level | QpiGNN | | SQR-GNN | | RQR$^{adj.}$-GNN | |
|---|---|---|---|---|---|---|---|
| | | PICP | MPIW | PICP | MPIW | PICP | MPIW |
| Feature Noise ($\sigma$) | 0.1 | 0.89 | 0.66 | 0.76 | 0.78 | 0.85 | 0.78 |
| | 0.2 | 0.90 | 0.80 | 0.65 | 0.69 | 0.85 | 0.78 |
| | 0.3 | 0.92 | 0.83 | 0.71 | 0.81 | 0.84 | 0.76 |
| Target Noise ($\sigma$) | 0.1 | 0.96 | 0.44 | 0.49 | 0.52 | 0.95 | 0.87 |
| | 0.2 | 0.99 | 0.71 | 0.95 | 0.84 | 0.97 | 1.05 |
| | 0.3 | 1.00 | 1.05 | 0.98 | 0.99 | 0.98 | 1.32 |
| Edge Dropout ($p$) | 0.2 | 0.84 | 0.21 | 0.53 | 0.44 | 0.86 | 0.80 |
| | 0.4 | 0.88 | 0.23 | 0.60 | 0.44 | 0.89 | 0.84 |
| | 0.6 | 0.91 | 0.21 | 0.82 | 0.50 | 0.89 | 0.84 |

*Table 3.* Exchangeability analysis on real-world datasets. We report average PICP and MPIW over 5 runs under three split types.

| Split Type | Education | | Election | | Income | | Unemploy. | | Twitch | |
|---|---|---|---|---|---|---|---|---|---|---|
| | PICP | MPIW | PICP | MPIW | PICP | MPIW | PICP | MPIW | PICP | MPIW |
| Random | 0.99 | 0.90 | 1.00 | 0.97 | 1.00 | 0.72 | 1.00 | 0.93 | 0.98 | 0.54 |
| Degree | 0.99 | 0.75 | 0.99 | 0.87 | 0.99 | 0.44 | 0.99 | 0.93 | 0.94 | 0.68 |
| Community | 0.99 | 0.54 | 0.99 | 0.86 | 1.00 | 0.46 | 0.97 | 0.84 | 0.99 | 0.49 |

| Split Type | Chameleon | | Crocodile | | Squirrel | | Anaheim | | Chicago | |
|---|---|---|---|---|---|---|---|---|---|---|
| | PICP | MPIW | PICP | MPIW | PICP | MPIW | PICP | MPIW | PICP | MPIW |
| Random | 0.98 | 0.40 | 1.00 | 0.54 | 0.99 | 0.47 | 0.99 | 0.74 | 0.99 | 0.60 |
| Degree | 0.97 | 0.57 | 1.00 | 0.72 | 0.97 | 0.45 | 0.95 | 0.58 | 0.96 | 0.46 |
| Community | 0.95 | 0.58 | 0.98 | 0.74 | 0.97 | 0.47 | 0.97 | 0.60 | 1.00 | 0.53 |

*Figure 4.* Radar plots of QpiGNN performance (PICP and MPIW) on synthetic graphs, demonstrating robustness to structural shifts.

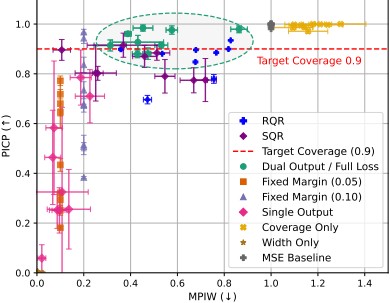

*Figure 5.* PICP–MPIW trade-off on nine synthetic datasets, comparing our ablation with SQR and RQR.

### 5.2.5. EXCHANGEABILITY UNDER SPLITS

Table 3 reports QpiGNN performance under three splits: *Random*, assigning nodes uniformly; *Degree*, separating nodes by degree to test generalization across low- and high-degree regions; and *Community*, splitting by communities to evaluate cross-community transfer. Results show that while *Random* splits yield stable coverage and widths, *Degree* splits often increase interval width (e.g., *Twitch*, MPIW = 0.68), and *Community* splits frequently produce narrower yet valid intervals (e.g., *Education*, 0.90 → 0.54). QpiGNN maintains coverage and adapts widths even under non-exchangeable splits, showing robustness to shifts.

### 5.2.6. COMPUTATIONAL EFFICIENCY

Appendix L compares efficiency, runtime, and complexity across baselines. QpiGNN balances accuracy and efficiency, with training cost comparable to lightweight baselines (e.g., SQR-GNN). All models share complexity $\mathcal{O}(Ed + Nd^2)$. MC Dropout and BayesianNN add overhead from sampling, while CF-GNN incurs the highest cost from calibration.

### 5.2.7. ABLATION STUDY

We examine the contributions of QpiGNN's architecture and loss design on nine synthetic datasets. Figure 5 presents

the PICP–MPIW trade-off, with results reported in Appendix N. *(1) Architecture:* The dual-head model with a learnable margin (green points) consistently outperforms fixed-margin and single-output variants, achieving superior calibration–compactness trade-offs. *(2) Loss:* The complete objective delivers the best overall performance (green dashed ellipse). Coverage-only training yields overly wide intervals, width-only training fails to calibrate, and pure MSE baseline collapses to trivial outputs, underscoring the importance of calibrated, uncertainty-aware losses.

### 5.2.8. HYPERPARAMETER SENSITIVITY

Figure 6 illustrates that at $1-\alpha = 0.90$, PICP remains stable ($\geq 0.96$) until $\lambda \approx 0.5$, after which both PICP and MPIW degrade rapidly, suggesting a critical tipping point. Bayesian optimization (Snoek et al., 2012) can be applied to select $\lambda_{\text{width}}$ for each dataset. As shown in Appendix E, optimal $\lambda$ values lie between 0.2 and 0.5, with synthetic datasets favoring stronger regularization. Overall, this range consistently yields good calibration–compactness trade-offs, while revealing that precise coverage control remains challenging for certain baselines. Additional analyses for the baselines and limitations are provided in Appendix D.

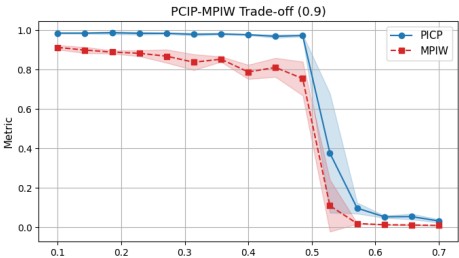

*Figure 6.* PICP-MPIW trade-off as a function of $\lambda_{\text{width}}$ at $1-\alpha = 0.90$, averaged over 5 runs (500 epochs) on the ER graph.

# 6. Discussion

QpiGNN's design yields strong empirical performance: by decoupling prediction and uncertainty, it avoids extra procedures and provides stable, calibrated estimates under noise and dependencies. Consistently, it produces tighter reliable intervals than baselines, demonstrating efficiency and robustness in line with UQ principles that prioritize reducing width when exceeding target coverage.

## 6.1. Constraints and Empirical Robustness

Theoretical guarantees of QpiGNN rely on mild assumptions that may be violated in real graphs due to heavy-tailed noise (Verma & Zhang, 2019; Jin et al., 2020), model bias (Pouplin et al., 2024), and structural redundancy (Tagasovska & Lopez-Paz, 2019). These factors represent practical violations of independence and symmetry, often limiting the reliability of UQ in graph data. Nevertheless, our results show that QpiGNN maintains calibrated and compact intervals across diverse settings—including noisy, sparse, and high-variance graphs—outperforming baselines on seven real datasets (Table 1). It further demonstrates resilience to feature and edge noise (Table 2) and sustains performance even under splits violating exchangeability (Table 3), highlighting that our method does not rely on strong assumptions. QpiGNN's limitations may be mitigated by incorporating coverage-aware objectives or robust training strategies to reduce model bias and handle heavy-tailed noise and structural redundancy. Moreover, the bounded-difference assumption may behave differently in dense or high-degree graphs, and clarifying this limitation is a direction for theoretical refinement.

## 6.2. Task Extensions

QpiGNN extends naturally beyond node regression. For graph-level regression, pooled embeddings $\mathbf{h}_G = \rho(\text{GNN}(\mathbf{X}, \mathcal{E}))$ are fed into two linear heads: $\hat{y}_G = \mathbf{W}_{\text{pred}}\mathbf{h}_G + b_{\text{pred}}$, $\hat{d}_G = \text{Softplus}(\mathbf{W}_{\text{diff}}\mathbf{h}_G + b_{\text{diff}})$, yielding prediction intervals $[\hat{y}_G - \hat{d}_G,\ \hat{y}_G + \hat{d}_G]$. For link prediction, edge embeddings (e.g., $\mathbf{h}_{uv} = \phi([\mathbf{h}_u \parallel \mathbf{h}_v \parallel \mathbf{h}_u \odot \mathbf{h}_v])$) are

processed in the same way, producing calibrated predictions $\hat{y}_{uv} \pm \hat{d}_{uv}$. While classification is not directly supported, the quantile-free dual-head design can be adapted via predictive sets or confidence margins.

# 7. Conclusion

We propose QpiGNN, a GNN-based framework for uncertainty quantification that estimates calibrated and compact prediction intervals for node-level regression. By combining a dual-head architecture with a quantile-free joint loss, QpiGNN enables end-to-end uncertainty estimation without requiring quantile-level inputs, resampling, or post-processing. Experiments on diverse graph datasets show that QpiGNN produces well-calibrated prediction intervals that remain robust to noise, non-exchangeability, and structural shifts. Ablation studies further confirm the contribution of the dual-head design and quantile-free joint loss on performance. Future work includes extending QpiGNN to broader graph applications requiring reliable UQ and improving its uncertainty modeling by disentangling aleatoric and epistemic components, enabling more informed decision-making in high-stakes settings. Another important direction is applying QpiGNN to dynamic or evolving graphs, where structural drift and temporal dependencies introduce additional uncertainties that remain unexplored.

## Acknowledgements

This work was supported by the National Research Foundation of Korea (NRF) grants funded by the Korea government, the Ministry of Education (MOE) (No. RS-2025-25422445), and the Ministry of Science and ICT (MSIT) (No. RS-2025-25435830).

## Impact Statement

This paper presents QpiGNN, a framework for uncertainty quantification in GNNs, whose goal is to advance the field of Machine Learning. Reliable uncertainty estimates are critical in high-stakes domains such as healthcare and criminal justice, where miscalibrated predictions can lead to harmful decisions. By providing calibrated and compact prediction intervals without requiring strong assumptions or costly post-processing, our work aims to improve the trustworthiness of GNN-based systems in real-world applications. There are many potential societal consequences of our work, none of which we feel must be specifically highlighted here.

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

# A. Training Algorithm for QpiGNN

---

**Algorithm 1** Training QpiGNN with Coverage–Width Joint Loss

---

**Input:** Graph $G = (\mathcal{V}, \mathcal{E})$, node features $\mathbf{X} \in \mathbb{R}^{N \times d}$, targets $\mathbf{y} \in \mathbb{R}^N$, coverage $1 - \alpha$, $\lambda_{\text{width}} \geq 0$, epochs $T$, learning rate $\eta$, weight decay $\omega$.

**Output:** Trained parameters $\boldsymbol{\theta}$.

**Procedure.**

1. Initialize $\boldsymbol{\theta}$ and optimizer $\texttt{Adam}(\boldsymbol{\theta}; \eta, \omega)$.

2. For $t = 1, \ldots, T$:

    (a) Compute embeddings:
    $$\mathbf{H}_1 \leftarrow \text{ReLU}(\text{SAGEConv}_1(\mathbf{X}, \mathcal{E})),$$
    $$\mathbf{H}_2 \leftarrow \text{ReLU}(\text{SAGEConv}_2(\mathbf{H}_1, \mathcal{E})).$$

    (b) Dual-head outputs:
    $$\hat{\mathbf{y}} \leftarrow \text{Linear}_{\text{pred}}(\mathbf{H}_2),$$
    $$\hat{\mathbf{d}} \leftarrow \text{softplus}(\text{Linear}_{\text{width}}(\mathbf{H}_2)).$$

    (c) Intervals: $\hat{\mathbf{y}}^{\text{low}} \leftarrow \hat{\mathbf{y}} - \hat{\mathbf{d}}, \hat{\mathbf{y}}^{\text{up}} \leftarrow \hat{\mathbf{y}} + \hat{\mathbf{d}}$.

    (d) Coverage and violation:
    $$\hat{c} := \frac{1}{|\mathcal{V}|} \sum_{v \in \mathcal{V}} \mathbb{I}\big(\hat{y}_v^{\text{low}} \leq y_v \leq \hat{y}_v^{\text{up}}\big),$$
    $$\hat{\ell}_{\text{viol}} := \frac{1}{|\mathcal{V}|} \sum_{v \in \mathcal{V}} \Big(|y_v - \hat{y}_v^{\text{low}}| \, \mathbb{I}[y_v < \hat{y}_v^{\text{low}}] + |y_v - \hat{y}_v^{\text{up}}| \, \mathbb{I}[y_v > \hat{y}_v^{\text{up}}]\Big).$$

    (e) Loss:
    $$\mathcal{L}_{\text{coverage}} := (\hat{c} - (1 - \alpha))^2 + \hat{\ell}_{\text{viol}},$$
    $$\mathcal{L}_{\text{width}} := \frac{1}{|\mathcal{V}|} \sum_{v \in \mathcal{V}} \big(\hat{y}_v^{\text{up}} - \hat{y}_v^{\text{low}}\big),$$
    $$\mathcal{L}_{\text{total}} := \mathcal{L}_{\text{coverage}} + \lambda_{\text{width}} \mathcal{L}_{\text{width}}.$$

    (f) Update: one step on $\nabla_{\boldsymbol{\theta}} \mathcal{L}_{\text{total}}$.

---

Algorithm 1 outlines the training process for QpiGNN, which learns to generate node-specific prediction intervals through a dual-head architecture: one head predicts the mean response, while the other estimates the interval width. The total loss $\mathcal{L}_{\text{total}}$ combines a coverage constraint $\mathcal{L}_{\text{coverage}}$, which penalizes incorrect interval coverage, and a width penalty $\mathcal{L}_{\text{width}}$, which encourages compact and informative intervals. The contribution of the width term is controlled by a regularization weight $\lambda_{\text{width}} \geq 0$, which governs the trade-off between calibration and interval tightness. By eliminating the need for explicit quantile estimation or conformal post-processing, QpiGNN enables fast, one-pass uncertainty quantification (UQ) that scales efficiently with graph size and structure.

# B. Applying QpiGNN Loss to an RQR-Style Formulation

Keeping the architecture fixed, we replaced the QpiGNN objective with the RQR loss and evaluated both variants across the 19 synthetic and real datasets in Table 4. This controlled setup allows us to isolate the effect of the loss function while keeping the encoder and training pipeline identical.

*Table 4.* Comparison using QpiGNN loss replaced by the RQR loss across 19 synthetic and real datasets (10 runs, target coverage 90%).

| Model | Basic | | Gaussian | | Uniform | | Outlier | | Edge | | BA | | ER | | Grid | | Tree | |
|---|---|---|---|---|---|---|---|---|---|---|---|---|---|---|---|---|---|---|
| | PCIP | MPIW | PCIP | MPIW | PCIP | MPIW | PCIP | MPIW | PCIP | MPIW | PCIP | MPIW | PCIP | MPIW | PCIP | MPIW | PCIP | MPIW |
| QpiGNN ($\lambda = 0.5$) | 0.89 | 0.30 | 0.92 | 0.55 | 0.88 | 0.43 | 0.89 | 0.47 | 0.94 | 0.39 | 0.98 | 0.49 | 0.98 | 0.63 | 0.98 | 0.87 | 0.96 | 0.39 |
| QpiGNN ($\lambda = optimal$) | 0.90 | 0.30 | 0.95 | 0.64 | 0.93 | 0.62 | 0.90 | 0.49 | 0.94 | 0.54 | 0.98 | 0.48 | 0.98 | 0.63 | 0.99 | 0.93 | 0.96 | 0.39 |
| QpiGNN w/ RQRLoss ($\lambda = 0.1$) | 0.92 | 0.84 | 0.93 | 0.61 | 0.90 | 0.71 | 0.94 | 0.46 | 0.93 | 0.85 | 0.19 | 0.85 | 0.50 | 0.83 | 0.87 | 0.62 | 0.27 | 0.81 |

| Model | Education | | Election | | Income | | Unemploy. | | Twitch | | Chameleon | | Crocodile | | Squirrel | | Anaheim | | Chicago | |
|---|---|---|---|---|---|---|---|---|---|---|---|---|---|---|---|---|---|---|---|---|
| | PCIP | MPIW | PCIP | MPIW | PCIP | MPIW | PCIP | MPIW | PCIP | MPIW | PCIP | MPIW | PCIP | MPIW | PCIP | MPIW | PCIP | MPIW | PCIP | MPIW |
| QpiGNN ($\lambda = 0.5$) | 0.99 | 0.57 | 0.98 | 0.77 | 0.99 | 0.41 | 1.00 | 0.74 | 0.59 | 0.08 | 0.51 | 0.03 | 0.92 | 0.08 | 0.73 | 0.07 | 0.92 | 0.39 | 0.97 | 0.36 |
| QpiGNN ($\lambda = optimal$) | 0.99 | 0.59 | 0.98 | 0.77 | 0.99 | 0.44 | 1.00 | 0.73 | 0.94 | 0.36 | 0.96 | 0.23 | 0.97 | 0.16 | 0.96 | 0.18 | 0.93 | 0.40 | 0.98 | 0.36 |
| QpiGNN w/ RQRLoss ($\lambda = 0.1$) | 0.90 | 0.53 | 0.89 | 0.52 | 0.90 | 0.37 | 0.84 | 0.46 | 0.95 | 0.72 | 0.94 | 0.30 | 0.97 | 0.24 | 0.92 | 0.25 | 0.83 | 0.44 | 0.82 | 0.41 |

Across the synthetic datasets, QpiGNN maintains near-target coverage with compact intervals, whereas the RQR-based variant often collapses, showing under-coverage (e.g., 0.19 on BA, 0.27 on Tree) and inflated intervals. A similar pattern appears on real-world graphs: QpiGNN provides reliable coverage with reasonable interval widths, while the RQR-style model shows degradation, including under-coverage and wider intervals on datasets such as Twitch, Chameleon, and Crocodile.

Overall, these results confirm that simply substituting the loss with an RQR formulation is insufficient for stable UQ on graphs. The proposed joint loss (Equation (6)) is crucial for achieving both coverage reliability and interval compactness under graph-dependent residual structures.

# C. Theoretical Foundations of QpiGNN

In this appendix, we provide theoretical insights and guarantees underpinning the design of QpiGNN. Specifically, we analyze the method along three key dimensions:

1. The convergence of empirical coverage to the desired target level $1 - \alpha$

2. The optimality of predicted interval widths under the coverage constraint

3. The convergence behavior of the proposed loss function under gradient-based optimization.

These results offer a foundational understanding of why QpiGNN is both statistically sound and practically effective for node-level UQ in graph-structured data.

## C.1. Coverage Consistency

We formally establish that the empirical coverage $\hat{c}$, defined as the proportion of true targets contained within the predicted intervals, converges to the target level $1 - \alpha$ as the number of nodes increases. This result provides theoretical justification for the calibration behavior observed in QpiGNN.

**Proposition C.1** (Asymptotic Coverage Consistency). *Let $\hat{y}_v$ and $\hat{d}_v > 0$ denote the predicted mean and interval half-width for node $v$, forming the prediction interval $[\hat{y}_v^{low}, \hat{y}_v^{up}] = [\hat{y}_v - \hat{d}_v, \ \hat{y}_v + \hat{d}_v]$. Suppose the following conditions hold:*

- *(A1) The noise $\varepsilon_v = y_v - f(x_v)$ is bounded and independent across nodes;*

- *(A2) The predicted mean and width satisfy $\hat{y}_v \xrightarrow{P} \mathbb{E}[y_v|x_v]$ and $\hat{d}_v \xrightarrow{P} d_v^*$, and the loss $\mathcal{L}_{total}$ is Lipschitz-continuous in the model parameters;*

- *(A3) Node embeddings $\mathbf{H}$ are sufficiently expressive and bounded in norm.*

*Then, as $N \to \infty$,*

$$\hat{c} := \mathbb{P}_{v\sim\mathcal{V}}\left(\hat{y}_v^{low} \leq y_v \leq \hat{y}_v^{up}\right) \xrightarrow{P} 1 - \alpha.$$

*Proof.* We verify the conditions of the Weak Law of Large Numbers (WLLN) (Penrose & Yukich, 2003; Gama & Ribeiro, 2019) for the sequence $\{Z_v\}_{v=1}^N$, where each $Z_v := \mathbb{I}[\hat{y}_v^{low} \leq y_v \leq \hat{y}_v^{up}]$ is:

- Bounded: $Z_v \in [0, 1]$ for all $v$,

- Mean-convergent: $\mathbb{E}[Z_v] \to 1 - \alpha$ as $N \to \infty$,

- Weakly dependent (assumed negligible or bounded via local message passing).

Under these conditions, the empirical mean $\hat{c} = \frac{1}{N} \sum_{v=1}^{N} Z_v$ satisfies the WLLN:

$$\hat{c} \xrightarrow{P} 1 - \alpha.$$

$\square$

**Finite-sample Concentration Bounds**  While the above result ensures asymptotic calibration, we now provide finite-sample guarantees that quantify the deviation of $\hat{c}$ from its expected value. These bounds show that QpiGNN maintains reliable coverage even with moderate graph sizes.

**Proposition C.2** (Hoeffding-type Bound (Hoeffding, 1994)). *If $Z_v := \mathbb{I}[\hat{y}_v^{low} \leq y_v \leq \hat{y}_v^{up}] \in [0, 1]$ are independent for $v = 1, \ldots, N$, then for any $\delta \in (0, 1)$, with probability at least $1 - \delta$,*

$$|\hat{c} - \mathbb{E}[\hat{c}]| \leq \sqrt{\frac{\log(2/\delta)}{2N}}.$$

**Proposition C.3** (McDiarmid-type Bound (McDiarmid, 1989)). *Let $\hat{c} = f(X_1, \ldots, X_N)$, where $X_v = (x_v, y_v)$, and suppose that changing a single sample $X_v$ alters $\hat{c}$ by at most $1/N$. Then for any $\epsilon > 0$,*

$$\mathbb{P}\left(|\hat{c} - \mathbb{E}[\hat{c}]| > \epsilon\right) \leq 2 \exp\left(-2N\epsilon^2\right).$$

These concentration bounds confirm that QpiGNN's empirical coverage remains close to its expected value with high probability, thus providing calibration guarantees under both asymptotic and finite-sample settings.

## C.2. Width Optimality under Coverage Constraints

We now theoretically justify why the objective used in QpiGNN encourages compact prediction intervals while satisfying the coverage requirement. We formalize this as a constrained optimization problem where the goal is to minimize the average width of prediction intervals subject to a minimum coverage level.

**Theorem C.4** (Width-Optimality under Coverage). *Let each $y_v$ follow a symmetric, continuous distribution centered at $\hat{y}_v = \mathbb{E}[y_v \mid x_v]$. Then, the solution to the optimization problem*

$$\min_{\{\hat{d}_v \geq 0\}} \quad \frac{1}{N} \sum_{v=1}^{N} 2\hat{d}_v \quad subject\ to \quad \mathbb{P}_{v \sim \mathcal{V}}\left(\hat{y}_v^{low} \leq y_v \leq \hat{y}_v^{up}\right) \geq 1 - \alpha$$

*is given by $\hat{d}_v^* = F_v^{-1}(1 - \alpha/2)$, where $F_v^{-1}(\cdot)$ denotes the conditional quantile function of $|y_v - \hat{y}_v|$.*

*Proof.* Given the symmetry and continuity of $y_v$ around $\hat{y}_v$, the most compact interval capturing mass $1 - \alpha$ is symmetric about the mean. By the definition of quantiles, the smallest such symmetric interval corresponds to the threshold $\hat{d}_v^*$ such that $\mathbb{P}(|y_v - \hat{y}_v| \leq \hat{d}_v^*) = 1 - \alpha$, which gives $\hat{d}_v^* = F_v^{-1}(1 - \alpha/2)$. This value achieves the minimal width $2\hat{d}_v^*$ while satisfying the global coverage constraint. $\square$

**Lagrangian Relaxation.**  In practice, the true quantile function $F_v^{-1}(\cdot)$ is unknown and intractable to estimate directly. Instead, QpiGNN minimizes the following surrogate loss:

$$\left(\mathbb{P}_{v \sim \mathcal{V}}\left(\hat{y}_v^{low} \leq y_v \leq \hat{y}_v^{up}\right) - (1 - \alpha)\right)^2 + \lambda_{\text{width}} \cdot \mathbb{E}_{v \sim \mathcal{V}}\left[\hat{y}_v^{up} - \hat{y}_v^{low}\right].$$

which serves as a soft Lagrangian penalty formulation (Franceschi et al., 2019). The hyperparameter $\lambda_{\text{width}}$ modulates the trade-off between satisfying the coverage constraint and minimizing interval width. When empirical coverage exceeds the target, the model reduces interval sizes; when coverage falls short, it implicitly expands the intervals by adjusting the predicted margins. This leads to adaptive and stable training dynamics without requiring explicit quantile estimation.

## C.3. Convergence of the Training Objective

We analyze the convergence behavior of a simplified version of the training loss used in QpiGNN, omitting the violation loss term $\hat{\ell}_{\text{viol}}$. The objective is defined as:

$$\mathcal{L}_{\text{total}}(\theta) = \left(\mathbb{P}_{v\sim\mathcal{V}}\left(\hat{y}_v^{\text{low}} \leq y_v \leq \hat{y}_v^{\text{up}}\right) - (1-\alpha)\right)^2 + \lambda_{\text{width}} \cdot \mathbb{E}_{v\sim\mathcal{V}}\left[\hat{y}_v^{\text{up}} - \hat{y}_v^{\text{low}}\right],$$

where $\theta$ denotes the trainable parameters. This loss consists of a quadratic coverage penalty and a linear interval width term, both of which are differentiable with respect to $\theta$.

**Proposition C.5** (Convergence to a Stationary Point). *Assume:*

*(A1) Each component of $\mathcal{L}_{total}$ is continuous and piecewise smooth;*

*(A2) The gradient norm is bounded: $\|\nabla_\theta \mathcal{L}_{total}(\theta)\| \leq G$;*

*(A3) The learning rate $\eta_t$ satisfies: $\eta_t \to 0$, $\sum_{t=1}^{\infty} \eta_t = \infty$, and $\sum_{t=1}^{\infty} \eta_t^2 < \infty$.*

*Then the sequence of iterates $\theta_t$ generated by stochastic gradient descent satisfies:*

$$\lim_{t\to\infty} \mathbb{E}[\|\nabla_\theta \mathcal{L}_{total}(\theta_t)\|] = 0.$$

*Proof.* The total loss $\mathcal{L}_{\text{total}}(\theta)$ is composed of a differentiable quadratic coverage term and a linear width penalty, both of which are locally Lipschitz and smooth with respect to $\theta$. As such, $\mathcal{L}_{\text{total}}$ satisfies the regularity conditions required by standard results in stochastic non-convex optimization (Ghadimi & Lan, 2013). Therefore, under assumptions (A1)–(A3), stochastic gradient descent with diminishing step sizes converges to a first-order stationary point in expectation. Similar convergence guarantees also hold for adaptive optimizers such as Adam, provided the gradients remain bounded (Kingma & Ba, 2015; Reddi et al., 2018). □

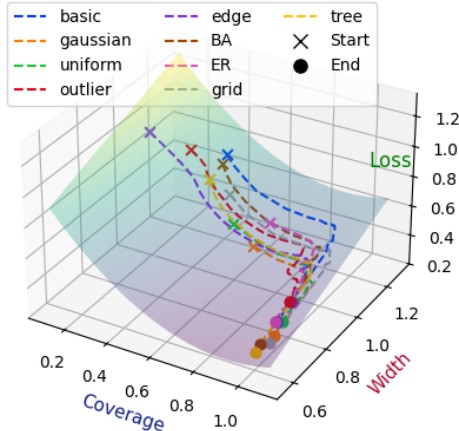
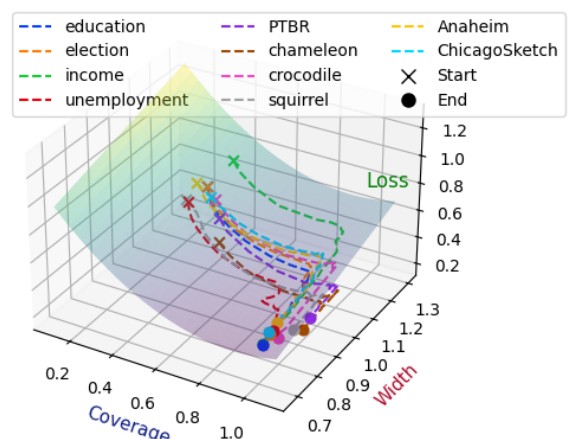

*(a)* Loss convergence trajectory on synthetic graph datasets.   *(b)* Loss convergence trajectory on real-world graph datasets.

*Figure 7.* 3D convergence trajectories in coverage–width–loss space. Each dashed line corresponds to a dataset, tracing optimization from initialization (✗) to convergence (●). The surfaces visualize the loss landscape, while the trajectories highlight stable descent toward calibrated and compact prediction intervals.

**Empirical verification.**   We empirically validate the convergence dynamics of QpiGNN through 3D trajectory visualizations on both synthetic and real-world graph datasets. Figure 7 plots training paths in the coverage–width–loss space, showing how the model jointly optimizes calibration and compactness over time. In both settings, training starts with high loss (marked by ✗) and proceeds toward a low-loss region (marked by ●) along a smooth descent trajectory.

On synthetic datasets (Figure 7a), the paths show a structured progression: the model first corrects coverage errors and then reduces interval width, eventually converging near the optimal coverage level $1 - \alpha = 0.9$ while minimizing width. These

trajectories confirm the effectiveness of the sequential optimization behavior embedded in our loss design. On real datasets (Figure 7b), the convergence paths are more varied due to structural heterogeneity and noise. Some datasets exhibit greater fluctuations in coverage during early training, but all converge toward the loss-minimizing surface. This robustness across dataset types highlights the adaptability and stability of QpiGNN's training process.

## C.4. Violation Loss and Theoretical Coverage Guarantees

Recall the total coverage-related loss is defined as $\mathcal{L}_{\text{coverage}} = (\hat{c} - (1 - \alpha))^2 + \hat{\ell}_{\text{viol}}$, where the empirical violation loss is

$$\hat{\ell}_{\text{viol}} := \frac{1}{N} \sum_{v=1}^{N} \left[ |y_v - \hat{y}_v^{\text{low}}| \cdot \mathbb{I}[y_v < \hat{y}_v^{\text{low}}] + |y_v - \hat{y}_v^{\text{up}}| \cdot \mathbb{I}[y_v > \hat{y}_v^{\text{up}}] \right].$$

This term penalizes not just whether a coverage violation occurs, but also by how far $y_v$ lies outside the predicted interval. It plays a key role in shaping the model's predictions during training, especially when $\hat{c} \ll 1 - \alpha$. However, note that $\hat{\ell}_{\text{viol}} \xrightarrow{P} 0$ as the model becomes well-calibrated, i.e., when

$$\mathbb{P}(y_v < \hat{y}_v^{\text{low}} \text{ or } y_v > \hat{y}_v^{\text{up}}) \to \alpha.$$

In this regime, the expected violation magnitude vanishes:

$$\mathbb{E}\left[ \hat{\ell}_{\text{viol}} \right] \leq \mathbb{E}\left[ |y_v - \hat{y}_v^{\text{low}}| \cdot \mathbb{I}[y_v < \hat{y}_v^{\text{low}}] + |y_v - \hat{y}_v^{\text{up}}| \cdot \mathbb{I}[y_v > \hat{y}_v^{\text{up}}] \right] \to 0.$$

Hence, $\hat{\ell}_{\text{viol}}$ is asymptotically negligible and does not interfere with the convergence of $\hat{c}$ to the target coverage level $1 - \alpha$. The proof of coverage consistency thus holds even in its absence.

## C.5. Robustness of Theoretical Guarantees under Practical Conditions

Our coverage consistency theorem assumes that the samples $X_v = (x_v, y_v)$ are i.i.d., and that model predictions converge under bounded noise. These assumptions, while standard, are idealized. In practical graph settings, particularly with GNNs, message passing introduces weak dependencies among node predictions due to shared neighborhood structures (Verma & Zhang, 2019; Jin et al., 2020).

Nonetheless, we observe that predictions $\hat{y}_v^{\text{low}}, \hat{y}_v^{\text{up}}$ still rely primarily on local features and limited-depth neighborhoods. As a result, these weak dependencies do not significantly impair the empirical convergence behavior of the coverage estimator $\hat{c}$. In particular, our empirical results demonstrate that QpiGNN maintains tight coverage and compact intervals even under node and edge perturbations, supporting the practical validity of the assumptions used in our finite-sample analysis. In fact, the bounded-difference property required by McDiarmid's inequality (McDiarmid, 1989) remains approximately satisfied:

$$\sup_{X_1,\ldots,X_N,X_v'} |\hat{c}(X_1, \ldots, X_v, \ldots, X_N) - \hat{c}(X_1, \ldots, X_v', \ldots, X_N)| \leq \frac{1}{N} + \delta_G,$$

where $\delta_G$ is a graph-dependent residual term that diminishes under localized message aggregation. Similar forms of concentration under dependency graphs and weak mixing conditions have been studied in, supporting the plausibility of our empirical findings.

*Remark* C.6. To build intuition, consider sparse Erdős–Rényi (ER) graphs with average degree $\mathcal{O}(\log N)$. In such graphs, the influence of a single node diminishes rapidly with distance. For example, in GraphSAGE with mean aggregation, each neighbor contributes at most $\mathcal{O}(1/\deg(v))$ per layer, yielding an overall effect of $\mathcal{O}(1/N)$. Over $k$ layers, the cumulative influence remains bounded by $\mathcal{O}(k/N)$. This intuition aligns with prior work showing that the WLLN can hold under weak dependence in graph processes (Gama & Ribeiro, 2019).

Empirically, QpiGNN maintains tight coverage and compact intervals under diverse perturbations, confirming that these assumptions hold in practice (see Section 5).

Moreover, while the theoretical analysis assumes symmetric conditional distributions (e.g., for width optimality), we observe that QpiGNN maintains valid coverage and produces compact intervals even under moderately skewed or heavy-tailed distributions. This suggests that the method is robust to deviations from theoretical assumptions in practice, consistent with observations in robust quantile regression literature (Tagasovska & Lopez-Paz, 2019; Pouplin et al., 2024).

*Table 5.* Hyperparameter sensitivity analysis of target-coverage baselines (SQR, RQR, CF-GNN) evaluated on 10 real datasets over 10 independent runs.

| Model (target coverage) | Education PCIP | Education MPIW | Election PCIP | Election MPIW | Income PCIP | Income MPIW | Unemploy. PCIP | Unemploy. MPIW | Twitch PCIP | Twitch MPIW | Chameleon PCIP | Chameleon MPIW | Crocodile PCIP | Crocodile MPIW | Squirrel PCIP | Squirrel MPIW | Anaheim PCIP | Anaheim MPIW | Chicago PCIP | Chicago MPIW |
|---|---|---|---|---|---|---|---|---|---|---|---|---|---|---|---|---|---|---|---|---|
| SQR (0.80) | 0.78 | 0.28 | 0.75 | 0.32 | 0.78 | 0.19 | 0.79 | 0.35 | 0.09 | 0.02 | 0.11 | 0.01 | 0.15 | 0.01 | 0.09 | 0.01 | 0.67 | 0.35 | 0.73 | 0.23 |
| SQR (0.85) | 0.82 | 0.30 | 0.78 | 0.36 | 0.83 | 0.21 | 0.83 | 0.38 | 0.12 | 0.02 | 0.11 | 0.01 | 0.16 | 0.01 | 0.10 | 0.01 | 0.72 | 0.38 | 0.76 | 0.24 |
| SQR (0.95) | 0.88 | 0.36 | 0.85 | 0.43 | 0.87 | 0.23 | 0.88 | 0.44 | 0.12 | 0.03 | 0.14 | 0.02 | 0.20 | 0.01 | 0.09 | 0.01 | 0.74 | 0.43 | 0.77 | 0.27 |
| RQR (0.80) | 0.78 | 0.37 | 0.77 | 0.40 | 0.78 | 0.26 | 0.76 | 0.35 | 0.62 | 0.42 | 0.67 | 0.09 | 0.72 | 0.03 | 0.68 | 0.10 | 0.65 | 0.22 | 0.72 | 0.30 |
| RQR (0.85) | 0.83 | 0.41 | 0.82 | 0.44 | 0.83 | 0.28 | 0.80 | 0.39 | 0.68 | 0.46 | 0.73 | 0.09 | 0.75 | 0.03 | 0.75 | 0.10 | 0.68 | 0.23 | 0.80 | 0.32 |
| RQR(0.95) | 0.94 | 0.65 | 0.93 | 0.60 | 0.94 | 0.42 | 0.90 | 0.51 | 0.83 | 0.56 | 0.92 | 0.17 | 0.86 | 0.05 | 0.94 | 0.19 | 0.83 | 0.62 | 0.91 | 0.51 |
| CF-GNN (0.80) | 0.80 | 2.66 | 0.80 | 0.99 | 0.80 | 2.45 | 0.80 | 2.28 | 0.81 | 2.77 | - | - | - | - | - | - | 0.81 | 2.55 | 0.80 | 2.44 |
| CF-GNN (0.85) | 0.85 | 3.21 | 0.85 | 1.07 | 0.85 | 2.63 | 0.85 | 2.79 | 0.85 | 3.00 | - | - | - | - | - | - | 0.86 | 3.42 | 0.85 | 2.89 |
| CF-GNN (0.95) | 0.95 | 5.10 | 0.95 | 1.44 | 0.95 | 4.48 | 0.95 | 4.77 | 0.95 | 7.91 | - | - | - | - | - | - | 0.96 | 5.65 | 0.95 | 4.78 |

*Table 6.* Optimal $\lambda_{\text{width}}$ and corresponding MPIW for each dataset obtained via Bayesian optimization.

| Synthetic Dataset | Best $\lambda_{\text{width}}$ | MPIW | Real Dataset | Best $\lambda_{\text{width}}$ | MPIW |
|---|---|---|---|---|---|
| Basic | 0.5000 | 0.39 | Education | 0.4898 | 0.44 |
| Gaussian | 0.4138 | 0.58 | Election | 0.5000 | 0.73 |
| Uniform | 0.4219 | 0.65 | Income | 0.2264 | 0.36 |
| Outlier | 0.4875 | 0.46 | Unemploy. | 0.1930 | 0.42 |
| Edge | 0.4876 | 0.48 | PTBR | 0.3761 | 0.21 |
| BA | 0.5000 | 0.65 | Chameleon | 0.2875 | 0.21 |
| ER | 0.5000 | 0.62 | Crocodile | 0.4871 | 0.39 |
| Grid | 0.3994 | 0.75 | Squirrel | 0.2875 | 0.21 |
| Tree | 0.4879 | 0.39 | Anaheim | 0.4871 | 0.40 |
| - | - | | Chicago | 0.5000 | 0.34 |

Overall, the purpose of Appendix B.5 is to demonstrate that concentration-based reasoning remains appropriate in the regimes where QpiGNN is evaluated, even though the strict i.i.d. and symmetry assumptions are only approximately satisfied in practical graph settings. A rigorous extension of our theoretical framework to formally incorporate weak dependency models—e.g., via mixing conditions or graph-dependent concentration inequalities—remains an important direction for future work.

# D. Additional Hyperparameter Sensitivity Results for Baselines

We further evaluated SQR, RQR, and CF-GNN under multiple target-coverage levels to assess whether these baselines can realize different coverage–width trade-offs (Table 5).

Although SQR adjusts its quantile inputs based on the specified target coverage, its achieved coverage consistently fell short of the desired levels across most datasets and collapsed entirely on non-homophilous graphs. RQR exhibited similar behavior: modifying the target coverage produced only minor changes in interval width, while the realized coverage remained largely unchanged and often far below the target.

These results reveal a limitation of current UQ methods for graph regression: similar to OpiGNN, RQR shows limited sensitivity to the penalty magnitude, indicating that coverage may be inherently difficult to control through this mechanism. We therefore highlight this limitation and point to it as an area requiring further investigation to achieve stricter control over target coverage.

# E. Dataset-Specific $\lambda_{\text{width}}$ Selection

To investigate how the trade-off between calibration and sharpness varies across tasks, we perform dataset-wise tuning of the width penalty coefficient $\lambda_{\text{width}}$ using Bayesian optimization. The results are summarized in Table 6 and visualized in Figure 8.

We observe substantial variation in the optimal $\lambda_{\text{width}}$ across datasets. Synthetic graphs typically favor higher values ($\geq 0.4$) to prevent overly conservative intervals in structured or low-noise settings (e.g., *Tree*, *Edge*). Real-world datasets show greater

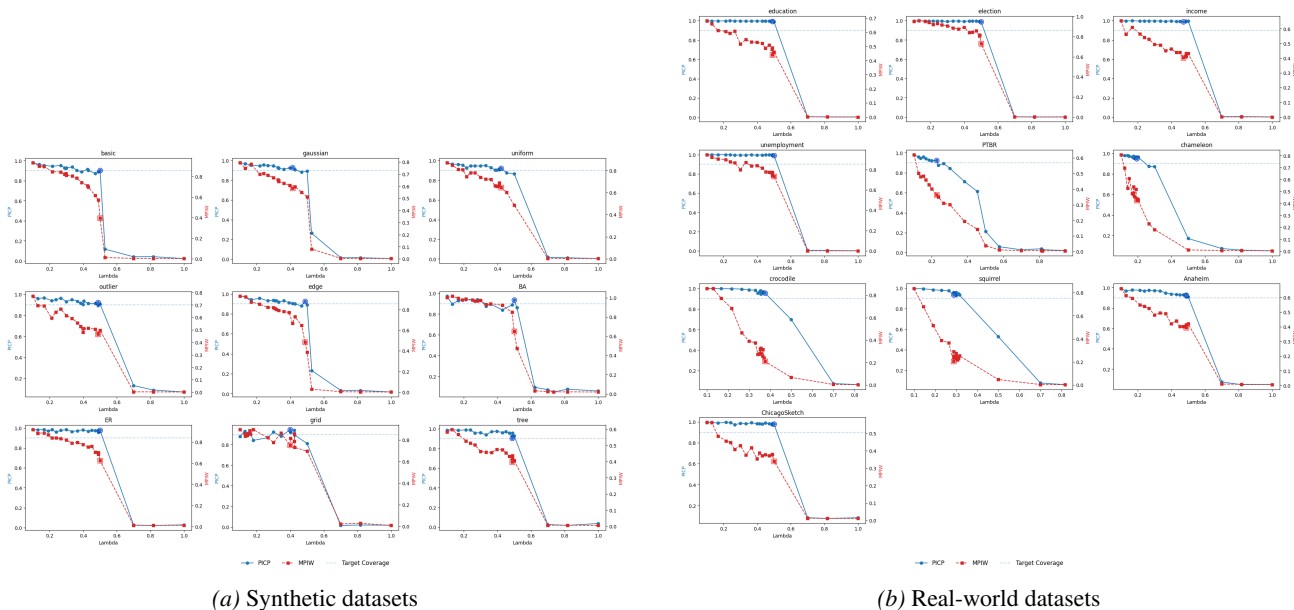

*(a)* Synthetic datasets                    *(b)* Real-world datasets

*Figure 8.* Optimal values of the width penalty coefficient $\lambda_{\text{width}}$ identified via Bayesian optimization for (a) synthetic and (b) real-world datasets. The results illustrate dataset-specific variability in the calibration–sharpness trade-off, supporting the need for adaptive regularization.

*Table 7.* Prediction interval performance on 19 synthetic and real datasets. Comparison of L1- and L2-based width penalties in QpiGNN (10 runs, target coverage 90%, $\lambda$ selected optimally).

| Penalty | Basic | | Gaussian | | Uniform | | Outlier | | Edge | | BA | | ER | | Grid | | Tree | |
|---|---|---|---|---|---|---|---|---|---|---|---|---|---|---|---|---|---|---|
| | PCIP | MPIW | PCIP | MPIW | PCIP | MPIW | PCIP | MPIW | PCIP | MPIW | PCIP | MPIW | PCIP | MPIW | PCIP | MPIW | PCIP | MPIW |
| L1 | 0.90 | 0.30 | 0.95 | 0.64 | 0.93 | 0.62 | 0.90 | 0.49 | 0.94 | 0.54 | 0.98 | 0.48 | 0.98 | 0.63 | 0.99 | 0.93 | 0.96 | 0.39 |
| L2 | 0.90 | 0.81 | 0.94 | 0.69 | 0.91 | 0.75 | 0.94 | 0.63 | 0.90 | 0.79 | 0.17 | 0.94 | 0.87 | 0.87 | 0.99 | 1.01 | 0.31 | 1.01 |

| Penalty | Education | | Election | | Income | | Unemploy. | | Twitch | | Chameleon | | Crocodile | | Squirrel | | Anaheim | | Chicago | |
|---|---|---|---|---|---|---|---|---|---|---|---|---|---|---|---|---|---|---|---|---|
| | PCIP | MPIW | PCIP | MPIW | PCIP | MPIW | PCIP | MPIW | PCIP | MPIW | PCIP | MPIW | PCIP | MPIW | PCIP | MPIW | PCIP | MPIW | PCIP | MPIW |
| L1 | 0.99 | 0.57 | 0.98 | 0.77 | 0.99 | 0.44 | 1.00 | 0.73 | 0.94 | 0.36 | 0.96 | 0.23 | 0.97 | 0.16 | 0.96 | 0.18 | 0.93 | 0.40 | 0.98 | 0.36 |
| L2 | 0.99 | 0.68 | 0.99 | 0.79 | 0.99 | 0.43 | 0.98 | 0.86 | 0.97 | 0.75 | 0.97 | 0.70 | 1.00 | 0.69 | 0.97 | 0.53 | 0.94 | 0.52 | 0.94 | 0.45 |

diversity: high-variability graphs (e.g., *Election*, *Crocodile*) prefer stronger regularization, while noisy or non-homophilous graphs (e.g., *Chameleon*, *Squirrel*) work better with smaller values (0.22–0.28). Notably, similar interval widths can arise from different $\lambda$ values depending on structural factors. These results highlight the need for dataset-specific tuning, and Bayesian optimization offers a principled way to achieve this.

## F. Comparison of L1 and L2 Width Penalties

We further validated this by replacing the L1 width penalty in Equation (6) with an L2 counterpart (Table 7). The L2 penalty consistently yielded much wider intervals—despite achieving similar or even overly conservative coverage—because it magnifies heavy-tailed residuals induced by heterophily, structural noise, and message passing. In contrast, the L1 penalty remained stable across all datasets and achieved the target coverage with substantially tighter intervals.

## G. Evaluation Datasets

We evaluate QpiGNN and all baselines on a diverse collection of graph-structured datasets, encompassing both synthetic and real-world settings. Table 8 summarizes key statistics across all datasets, including graph size, feature dimensions, and target distribution.

*Table 8.* Statistics of all datasets used in the experiments, including synthetic, real-world, and graph-structured domains. Each entry lists the number of nodes, edges, and input features.

| | Dataset | # Nodes | # Edges | # Features | | Dataset | # Nodes | # Edges | # Features |
|---|---|---|---|---|---|---|---|---|---|
| **Synthetic** | Basic | 1,000 | 2,000 | 5 | **Real** | Education | 3,234 | 12,717 | 6 |
| | Gaussian | 1,000 | 2,000 | 5 | | Election | 3,234 | 12,717 | 6 |
| | Uniform | 1,000 | 2,000 | 5 | | Income | 3,234 | 12,717 | 6 |
| | Outlier | 1,000 | 2,000 | 5 | | Unemploy. | 3,234 | 12,717 | 6 |
| | Edge | 1,000 | 2,000 | 5 | | Twitch | 1,912 | 31,299 | 3,169 |
| | BA | 1,000 | ~2,994 | 5 | | Chameleon | 2,277 | 36,101 | 3,132 |
| | ER | 1,000 | ~5,000 | 5 | | Crocodile | 11,631 | 180,020 | 13,183 |
| | Grid | 900 | 1,740 | 5 | | Squirrel | 5,201 | 217,073 | 3,148 |
| | Tree | 1,000 | 998 | 5 | | Anaheim | 914 | 3,638 | 4 |
| | - | | - | | | Chicago | 2,176 | 14,961 | 4 |

**Synthetic Datasets.** We generate nine synthetic datasets to assess controlled uncertainty estimation behavior under varying structural and statistical conditions. These datasets include grid, chain, tree, and random graphs, each with distinct patterns of homophily, noise level, and target function. Node features and targets are synthetically generated based on predefined functional relationships (e.g., nonlinear or spatial functions with added noise). The generation scripts and documentation are publicly available on GitHub[7]. These datasets enable controlled ablation and stress testing of uncertainty quantification performance.

**Real-World Datasets.** We also evaluate on ten publicly available real-world datasets from diverse domains, covering social networks, geographic data, knowledge graphs, and transportation systems.

- **U.S. County-Level Datasets (Education, Election, Income, Unemployment)** are constructed from county-level U.S. maps, using adjacency information derived from geographic boundaries. Node attributes include socioeconomic indicators, and targets reflect either vote shares or demographic statistics. The base topology and election outcomes were obtained from an open GitHub repository[8], while additional attributes were sourced from the U.S. Department of Agriculture Economic Research Service[9].

- **Wikipedia & Twitch Graphs (Chameleon, Squirrel, Crocodile, PTBR)** were collected from the MUSAE project (Rozemberczki et al., 2021), which provides temporal and social graphs annotated with node features and continuous targets. These datasets are widely used for benchmarking node regression in non-homophilous graphs[10].

- **Transportation Networks (Anaheim, Chicago)** model urban road networks as graphs, with nodes corresponding to intersections and edges to road segments. Node features include traffic-related metrics, and the targets correspond to flow estimates or congestion levels. These datasets were obtained from the Transportation Networks for Research repository[11].

These datasets span a wide range of graph topologies, feature modalities, and target distributions, providing a robust testbed for evaluating node-level prediction interval quality across domains.

# H. Evaluation Metrics

To evaluate the quality of UQ, we consider a comprehensive set of metrics that collectively measure two core aspects of prediction intervals: *reliability* (calibration) and *informativeness* (compactness). These metrics enable a nuanced analysis of model performance beyond simple point prediction accuracy.

There is a natural tension between reliability (PICP) and informativeness (MPIW/sharpness): increasing interval width improves coverage but reduces precision. A robust UQ model should strike a balance between the two. Metrics like WS and

---

[7] https://github.com/sybeam27/QpiGNN
[8] https://github.com/tonmcg/
[9] https://www.ers.usda.gov/data-products/county-level-data-sets/
[10] https://github.com/benedekrozemberczki/MUSAE
[11] https://github.com/bstabler/TransportationNetworks

CWC explicitly model this trade-off, with CWC particularly effective in safety-critical domains where undercoverage must be avoided at all costs.

**Prediction Interval Coverage Probability (PICP)**   PICP (Rana et al., 2015) evaluates whether the model's prediction intervals capture the true target values at the desired rate. It is the proportion of samples whose ground truth values fall within the corresponding predicted intervals:

$$\text{PICP} = \frac{1}{N} \sum_{v=1}^{N} \mathbb{I}\left[\hat{y}_v^{\text{low}} \leq y_v \leq \hat{y}_v^{\text{up}}\right].$$

A well-calibrated model with target coverage $1 - \alpha$ should achieve PICP close to that value. However, high PICP alone does not guarantee high-quality intervals if the width is excessive.

**Mean Prediction Interval Width (MPIW)**   MPIW (Khosravi et al., 2010a) measures the average width of the predicted intervals:

$$\text{MPIW} = \frac{1}{N} \sum_{v=1}^{N} (\hat{y}_v^{\text{up}} - \hat{y}_v^{\text{low}}).$$

Smaller MPIW indicates tighter (sharper) intervals, which are generally more informative. However, overly narrow intervals risk missing the true target and reducing PICP. Thus, MPIW should be interpreted jointly with PICP.

**Mean Prediction Error (MPE)**   MPE measures the average deviation between the center of the predicted interval and the ground truth:

$$\text{MPE} = \frac{1}{N} \sum_{v=1}^{N} \left| \frac{\hat{y}_v^{\text{low}} + \hat{y}_v^{\text{up}}}{2} - y_v \right|.$$

This evaluates whether the intervals are centered correctly. A model with good coverage and sharpness may still perform poorly if it consistently shifts intervals away from the true value.

**Sharpness**   Sharpness (Gneiting & Raftery, 2007) refers to the concentration of prediction intervals and penalizes unnecessarily wide intervals:

$$\text{Sharpness} = \frac{1}{N} \sum_{v=1}^{N} \left(\hat{y}_v^{\text{up}} - \hat{y}_v^{\text{low}}\right)^2.$$

It is a stricter version of MPIW that emphasizes outliers by squaring the width. A well-calibrated model should aim for minimal sharpness under valid PICP.

**Winkler Score (WS)**   WS (Winkler, 1994) is a proper scoring rule that evaluates both the width of the interval and whether it covers the true target. It imposes a linear penalty on width and an additional penalty on uncovered samples:

$$\text{WS} = \frac{1}{N} \sum_{v=1}^{N} \left[ \left(\hat{y}_v^{\text{up}} - \hat{y}_v^{\text{low}}\right) + \frac{2}{\alpha} \cdot \max(0, \ \hat{y}_v^{\text{low}} - y_v, \ y_v - \hat{y}_v^{\text{up}}) \right].$$

WS is interpretable, sensitive to both undercoverage and over-conservativeness, and widely used in applied settings.

**Combinational Coverage Width-based Criterion (CWC)**   CWC (Khosravi et al., 2010b) balances calibration and sharpness with an asymmetric penalty. It penalizes undercoverage exponentially, making it highly sensitive to violations of the target coverage:

$$\text{CWC} = \text{NMPIW} \cdot \left(1 + \gamma \cdot e^{-\eta(\text{PICP}-\mu)}\right),$$

where $\mu = 1 - \alpha$, $\gamma = 1$, and $\eta = 10$. When PICP falls below $\mu$, the exponential term grows rapidly, strongly penalizing the score.

**Normalized Mean Prediction Interval Width (NMPIW)**    NMPIW normalizes MPIW to make interval widths comparable across datasets with different label ranges:

$$\text{NMPIW} = \frac{1}{N} \sum_{v=1}^{N} \frac{\hat{y}_v^{\text{up}} - \hat{y}_v^{\text{low}}}{y_{\text{max}} - y_{\text{min}}}.$$

It is especially useful for cross-dataset comparisons and is a key component of CWC.

## I. Compared Baseline Models

*Table 9.* Summary of baseline models in terms of strengths, limitations, and computational complexity.

| Model | Strengths | Limitations | Comp. Cost |
|---|---|---|---|
| **SQR** | Fast and simple; enables arbitrary quantile estimation without sampling; compact implementation | No coverage guarantee; intervals may cross; sensitive to quantile choices | Low |
| **RQR** | Predicts both bounds jointly; soft ordering constraint avoids interval crossing; promotes compact intervals | Requires tuning of penalties; prone to over-smoothing when used with GNNs | Moderate |
| **CF-GNN** | Post-hoc calibrated intervals with valid marginal coverage; model-agnostic; flexible scoring rules | Needs separate calibration set; assumes exchangeability; less adaptive to node-level uncertainty | Moderate |
| **BayesianNN** | Theoretically grounded; captures both epistemic and aleatoric uncertainty | High computational cost due to sampling; slow convergence; less scalable | High |
| **MC Dropout** | Easy to integrate; empirically effective in practice; low overhead at training time | Multiple forward passes at inference; unstable under high variance; approximate posterior | Moderate |

We compare QpiGNN against five representative baseline models that span distinct paradigms of uncertainty quantification: quantile regression, Bayesian approximation, and conformal prediction. All baselines are adapted to graph-based settings using the same GNN backbone (except CF-GNN$^{opt}$, which includes its own calibration-specific components).

**Simultaneous Quantile Regression (SQR)**    SQR (Tagasovska & Lopez-Paz, 2019) models conditional quantiles by treating the quantile level $\tau \in (0, 1)$ as an input feature. To construct prediction intervals, the model is evaluated twice at $\tau_{\text{low}} = \alpha/2$ and $\tau_{\text{up}} = 1 - \alpha/2$, where $\alpha$ is the miscoverage level. The model is trained using the standard quantile (pinball) loss. While SQR allows simultaneous learning of multiple quantiles through randomized sampling over $\tau$, the model requires explicitly specified quantile levels and may suffer from quantile crossing (Zhou et al., 2020a).

**Relaxed Quantile Regression (RQR)**    RQR (Pouplin et al., 2024) directly predicts both lower and upper bounds of prediction intervals using a shared architecture. The training objective incorporates a composite loss that balances calibration and compactness:

$$\mathcal{L}_{\text{RQR}} = \mathcal{L}_{\text{RQR-W}} + \gamma_{\text{order}} \cdot \text{ReLU}(\hat{\mathbf{y}}^{\text{low}} - \hat{\mathbf{y}}^{\text{up}}),$$

where $\mathcal{L}_{\text{RQR-W}}$ penalizes miscoverage and excessive width, and the ReLU term enforces a soft constraint to avoid interval crossing. $\gamma_{\text{order}} \geq 0$ is a hyperparameter. This ordering penalty is *not included in the original formulation* but added here to improve interval validity in GNN-based tasks. Despite its design, RQR tends to produce overly smooth and wide intervals when applied to GNNs, due to their intrinsic neighborhood averaging and representation homogeneity (Rusch et al., 2023).

**Bayesian Neural Networks (BayesianNN)**    BayesianNN (Kendall & Gal, 2017) models uncertainty via posterior distributions over network weights. During inference, multiple stochastic forward passes are performed to estimate the predictive mean $\mu$ and standard deviation $\sigma$, forming prediction intervals as:

$$[\mu - t \cdot \sigma, \ \mu + t \cdot \sigma], \quad t \text{ chosen to match the desired confidence level.}$$

This method captures both epistemic and aleatoric uncertainty but is computationally expensive and often slow to converge.

*Table 10.* Comparison of Mean Prediction Error (MPE) and Combinational Coverage Width-based Criterion (CWC) on synthetic and real datasets. Lower values indicate better performance for both metrics. The best result for each dataset is highlighted in **bold**, and the second-best is underlined.

| | Model | Basic | | Gaussian | | Uniform | | Outlier | | Edge | | BA | | ER | | Grid | | Tree | |
|---|---|---|---|---|---|---|---|---|---|---|---|---|---|---|---|---|---|---|---|
| | | MPE | CWC | MPE | CWC | MPE | CWC | MPE | CWC | MPE | CWC | MPE | CWC | MPE | CWC | MPE | CWC | MPE | CWC |
| Synthetic | SQR-GNN | 0.09 | 1.01 | **0.13** | 1.32 | 0.13 | 1.27 | **0.05** | **0.20** | **0.08** | **0.64** | 0.25 | 3.20 | 0.21 | 4.27 | 0.17 | 2.87 | **0.08** | 1.38 |
| | RQR$^{adj.}$-GNN | 0.25 | 1.66 | 0.14 | 1.34 | 0.17 | 1.48 | 0.11 | 0.71 | 0.25 | 1.43 | 0.28 | 3.41 | 0.25 | 1.77 | 0.17 | 3.87 | 0.24 | 2.14 |
| | BayesianNN | 0.39 | 4.15 | 0.31 | 4.63 | 0.35 | 4.39 | 0.09 | 3.94 | 0.42 | 4.22 | 0.40 | 4.43 | 0.28 | 4.34 | 0.30 | 4.53 | 0.29 | 4.79 |
| | MC dropout | 0.28 | 71.71 | 0.15 | 34.7 | 0.21 | 60.18 | 0.07 | 8.18 | 0.27 | 58.17 | 0.24 | 47.61 | 0.28 | 157.92 | **0.15** | 55.25 | 0.27 | 85.17 |
| | CF-GNN | 0.57 | 3.26 | 0.74 | 2.50 | 0.85 | 4.05 | 0.54 | 0.67 | 0.50 | 3.01 | 17.67 | 59.12 | 4.62 | 14.57 | 0.95 | 11.82 | 0.26 | 1.80 |
| | QpiGNN ($\lambda^{0.5}$) | **0.08** | 0.63 | **0.13** | 1.15 | 0.11 | **1.02** | 0.18 | 0.99 | 0.10 | 0.65 | **0.11** | 0.75 | 0.15 | 0.97 | 0.16 | **1.39** | **0.08** | 0.70 |
| | QpiGNN ($\lambda^{0.1}$) | 0.25 | 1.33 | 0.15 | 1.36 | 0.18 | 1.35 | 0.23 | 1.04 | 0.26 | 1.34 | 0.32 | 1.56 | 0.24 | 1.36 | 0.17 | 1.57 | 0.09 | 0.94 |
| | QpiGNN ($\lambda^{opt.}$) | **0.08** | **0.62** | 0.14 | 1.18 | 0.14 | 1.16 | 0.19 | 1.00 | 0.14 | 0.91 | **0.11** | **0.74** | 0.15 | 0.97 | 0.15 | 1.44 | **0.08** | 0.70 |

| | Model | Education | | Election | | Income | | Unemploy. | | Twitch | | Chameleon | | Crocodile | | Squirrel | | Anaheim | | Chicago | |
|---|---|---|---|---|---|---|---|---|---|---|---|---|---|---|---|---|---|---|---|---|---|
| | | MPE | CWC | MPE | CWC | MPE | CWC | MPE | CWC | MPE | CWC | MPE | CWC | MPE | CWC | MPE | CWC | MPE | CWC | MPE | CWC |
| Real | SQR-GNN | **0.09** | **0.55** | **0.12** | 1.02 | **0.06** | 0.56 | **0.08** | 1.01 | **0.03** | 20.77 | **0.02** | 2.39 | **0.01** | 0.75 | **0.02** | 10.73 | **0.10** | 1.06 | **0.06** | 0.58 |
| | RQR$^{adj.}$-GNN | 0.15 | 0.87 | 0.14 | 1.17 | 0.10 | 0.74 | 0.09 | 0.99 | 0.10 | 1.30 | 0.07 | 0.59 | 0.04 | 0.27 | 0.07 | 0.31 | 0.20 | 1.76 | 0.10 | 0.77 |
| | BayesianNN | 0.16 | 3.11 | 0.37 | 4.25 | 0.14 | 3.96 | 0.16 | 5.15 | 0.18 | 6.88 | 0.08 | 6.24 | 0.08 | 4.53 | 0.08 | 4.06 | 0.15 | 5.36 | 0.10 | 4.27 |
| | MC dropout | 0.15 | 64.77 | 0.14 | 35.59 | 0.11 | 57.30 | 0.09 | 44.11 | 0.12 | 44.28 | 0.04 | 26.19 | 0.03 | 12.55 | 0.04 | 27.65 | 0.18 | 58.96 | 0.10 | 82.75 |
| | CF-GNN | 0.72 | 0.81 | 0.26 | 1.34 | 0.75 | 0.97 | 0.75 | 0.87 | 0.77 | 1.26 | - | - | - | - | - | - | 0.89 | 1.37 | 0.75 | 0.89 |
| | CF-GNN (opt.) | 0.76 | 0.98 | 0.22 | 1.03 | 0.71 | 0.81 | 0.67 | **0.73** | 0.57 | **0.75** | - | - | - | - | - | - | 1.00 | 1.37 | 0.67 | 0.65 |
| | QpiGNN ($\lambda^{0.5}$) | 0.11 | 0.61 | 0.15 | 1.17 | 0.07 | **0.56** | 0.18 | 1.29 | 0.04 | 3.79 | 0.03 | 3.56 | 0.02 | **0.16** | 0.03 | 0.58 | 0.12 | 0.95 | 0.07 | **0.55** |
| | QpiGNN ($\lambda^{0.1}$) | 0.23 | 0.96 | 0.17 | 1.39 | 0.13 | 0.96 | 0.23 | 1.61 | 0.11 | 1.28 | 0.09 | 0.91 | 0.06 | 0.80 | 0.08 | 0.66 | 0.19 | 1.37 | 0.11 | 0.87 |
| | QpiGNN ($\lambda^{opt.}$) | 0.12 | 0.64 | 0.14 | 1.15 | 0.07 | 0.59 | 0.18 | 1.28 | 0.08 | 1.00 | 0.06 | **0.55** | 0.03 | 0.26 | 0.04 | **0.29** | 0.12 | **0.94** | 0.07 | **0.55** |

**Monte Carlo Dropout (MC dropout)** MC dropout (Gal & Ghahramani, 2016) offers an approximate Bayesian alternative by retaining dropout during inference. Similar to BayesianNN, the model performs multiple forward passes to compute predictive mean and variance. It is simpler to implement and more scalable, but its uncertainty estimates can be unstable in high-variance regimes.

**Conformalized GNN (CF-GNN)** CF-GNN (Huang et al., 2023) separates prediction and calibration by first training a base GNN regressor $\hat{\mathbf{y}}$, and then post-calibrating the intervals using conformal prediction (CP) methods. The calibrated prediction intervals is:

$$[\hat{\mathbf{y}} - \hat{q}, \ \hat{\mathbf{y}} + \hat{q}],$$

where $\hat{q}$ is the $(1 - \alpha)$-quantile of residuals on a held-out calibration set. CF-GNN is model-agnostic and guarantees valid marginal coverage under the exchangeability assumption. However, it requires an additional calibration dataset and may be sensitive to distribution shifts. We use the official PyTorch-Geometric implementation[12] provided by the authors.

Table 9 summarizes the comparative properties of the five baselines. These models reflect a wide range of UQ approaches—quantile-based, Bayesian, and CP—and highlight the trade-offs in coverage validity, interval compactness, computational efficiency, and graph-awareness. For fairness, we implement all models using the same GNN encoder as QpiGNN, except CF-GNN$^{opt}$, which uses its official architecture and calibration setup.

## J. Supplementary Metric Results

In addition to the primary evaluation metrics—PICP and MPIW—we report results on four supplementary UQ metrics to provide a more comprehensive evaluation. Table 10 presents results on MPE and CWC. MPE captures the alignment between the center of each predicted interval and the corresponding ground truth value, serving as a proxy for point-prediction accuracy. CWC combines normalized interval width with an exponential penalty on undercoverage, allowing us to assess how well a model balances compactness with reliability. A lower CWC indicates better trade-off handling between sharpness and valid coverage. Table 11 reports results on Sharpness and WS, both of which focus on interval compactness and informativeness. Sharpness penalizes overly wide intervals through a quadratic term, while WS additionally incorporates a coverage-sensitive penalty, making it particularly useful for evaluating practical utility. Models like SQR-GNN and MC

---

[12]Source code: https://github.com/snap-stanford/conformalized-gnn

*Table 11.* Comparison of Sharpness and Winkler Score (WS) on synthetic and real datasets. Lower values indicate better performance for both metrics. The best result for each dataset is highlighted in **bold**, and the second-best is underlined.

| Model | Basic Sharpness | WS | Gaussian Sharpness | WS | Uniform Sharpness | WS | Outlier Sharpness | WS | Edge Sharpness | WS | BA Sharpness | WS | ER Sharpness | WS | Grid Sharpness | WS | Tree Sharpness | WS |
|---|---|---|---|---|---|---|---|---|---|---|---|---|---|---|---|---|---|---|
| SQR-GNN | 0.14 | 0.33 | 0.26 | **0.51** | 0.27 | 0.52 | 0.01 | 0.12 | 0.13 | **0.32** | 0.54 | 0.73 | 0.38 | 0.61 | 0.29 | 0.55 | **0.07** | **0.27** |
| RQR$^{adj.}$-GNN | 0.67 | 0.82 | 0.29 | 0.54 | 0.46 | 0.68 | 0.13 | 0.37 | 0.68 | 0.84 | 0.57 | 0.76 | 0.60 | 0.78 | 0.25 | 0.52 | 0.46 | 0.68 |
| BayesianNN | 9.08 | 3.01 | 8.92 | 2.98 | 9.02 | 3.00 | 8.72 | 2.95 | 9.37 | 3.06 | 9.50 | 3.08 | 9.09 | 3.01 | 9.11 | 3.01 | 9.02 | 3.00 |
| MC dropout | **0.10** | 0.47 | **0.04** | 0.27 | **0.07** | **0.36** | **0.00** | 0.45 | 0.09 | 0.45 | **0.07** | **0.41** | 0.07 | 0.41 | **0.03** | 0.25 | 0.04 | 0.38 |
| CF-GNN | 4.79 | 1.91 | 9.47 | 2.92 | 10.49 | 3.07 | 4.42 | 2.01 | 3.79 | 1.78 | 10514.81 | 68.69 | 638.45 | 17.26 | 14.91 | 3.19 | 1.31 | 0.98 |
| QpiGNN ($\lambda^{0.5}$) | **0.10** | **0.30** | 0.31 | 0.55 | 0.19 | 0.44 | 0.23 | 0.48 | 0.17 | 0.39 | 0.31 | 0.49 | 0.42 | 0.63 | 0.76 | 0.87 | 0.15 | 0.39 |
| QpiGNN ($\lambda^{0.1}$) | 0.86 | 0.93 | 0.71 | 0.84 | 0.79 | 0.89 | 0.56 | 0.75 | 0.94 | 0.97 | 1.03 | 1.01 | 0.85 | 0.92 | 0.95 | 0.98 | 0.35 | 0.59 |
| QpiGNN ($\lambda^{opt.}$) | **0.10** | 0.31 | 0.42 | 0.65 | 0.39 | 0.62 | 0.25 | 0.49 | 0.30 | 0.54 | 0.30 | 0.48 | 0.42 | 0.63 | 0.87 | 0.93 | 0.15 | 0.39 |

(Left margin label: Synthetic)

| Model | Education Sharpness | WS | Election Sharpness | WS | Income Sharpness | WS | Unemploy. Sharpness | WS | Twitch Sharpness | WS | Chameleon Sharpness | WS | Crocodile Sharpness | WS | Squirrel Sharpness | WS | Anaheim Sharpness | WS | Chicago Sharpness | WS |
|---|---|---|---|---|---|---|---|---|---|---|---|---|---|---|---|---|---|---|---|---|
| SQR-GNN | 0.11 | 0.33 | 0.22 | 0.48 | 0.05 | 0.23 | 0.11 | 0.34 | **0.00** | **0.05** | **0.00** | **0.03** | **0.00** | **0.01** | **0.00** | **0.03** | 0.12 | 0.32 | 0.05 | 0.22 |
| RQR$^{adj.}$-GNN | 0.24 | 0.50 | 0.29 | 0.55 | 0.13 | 0.37 | 0.15 | 0.39 | 0.18 | 0.42 | 0.02 | 0.16 | 0.01 | 0.09 | 0.02 | 0.16 | 0.25 | 0.51 | 0.09 | 0.31 |
| BayesianNN | 8.78 | 2.96 | 8.93 | 2.98 | 8.83 | 2.97 | 8.87 | 2.98 | 9.44 | 3.07 | 8.73 | 2.95 | 9.45 | 3.07 | 8.85 | 2.97 | 8.70 | 2.94 | 8.96 | 2.99 |
| MC dropout | **0.01** | **0.14** | **0.03** | **0.23** | **0.01** | **0.11** | **0.01** | **0.13** | 0.02 | 0.15 | **0.00** | 0.04 | **0.00** | 0.02 | **0.00** | 0.04 | **0.01** | **0.14** | **0.01** | **0.09** |
| CF-GNN | 8.78 | 2.83 | 1.25 | 1.10 | 14.32 | 3.55 | 10.77 | 3.24 | 14.33 | 3.56 | - | - | - | - | - | - | 11.49 | 3.28 | 11.11 | 3.18 |
| CF-GNN (opt.) | 11.82 | 3.14 | 0.90 | 0.96 | 8.63 | 2.97 | 6.88 | 2.69 | 5.58 | 2.38 | - | - | - | - | - | - | 8.17 | 2.86 | 5.56 | 2.31 |
| QpiGNN ($\lambda^{0.5}$) | 0.35 | 0.57 | 0.62 | 0.78 | 0.19 | 0.41 | 0.56 | 0.74 | 0.01 | 0.10 | **0.00** | 0.05 | 0.01 | 0.08 | 0.01 | 0.08 | 0.18 | 0.40 | 0.15 | 0.36 |
| QpiGNN ($\lambda^{0.1}$) | 0.80 | 0.90 | 0.93 | 0.97 | 0.52 | 0.72 | 0.87 | 0.93 | 0.32 | 0.54 | 0.18 | 0.41 | 0.32 | 0.54 | 0.24 | 0.47 | 0.56 | 0.74 | 0.37 | 0.60 |
| QpiGNN ($\lambda^{opt.}$) | 0.38 | 0.59 | 0.61 | 0.78 | 0.22 | 0.44 | 0.55 | 0.73 | 0.15 | 0.37 | 0.06 | 0.23 | 0.03 | 0.16 | 0.04 | 0.19 | 0.18 | 0.40 | 0.16 | 0.36 |

(Left margin label: Real)

dropout achieve low sharpness and WS, indicating tight intervals—but often at the expense of undercoverage.

As discussed in Section 5, such models fail to meet the target coverage threshold (e.g., $1 - \alpha = 0.9$) as shown in Table 1, thereby exposing a critical trade-off between reliability and precision. These additional metrics thus offer a more nuanced view of model performance, enabling a clearer understanding of uncertainty behavior across diverse graph datasets. They also help identify whether a method's compact intervals are meaningfully calibrated or merely overconfident.

# K. Qualitative Analysis

To complement our quantitative evaluation, we present a qualitative comparison of prediction intervals across all models. Figure 9 shows visualizations for nine synthetic datasets, while Figure 10 displays corresponding results for ten real-world graph datasets.

QpiGNN achieves a favorable trade-off between calibration and compactness. On the Tree dataset, SQR and MC Dropout produce narrow intervals that miss several true targets—demonstrating undercoverage—while RQR generates globally smooth but overly wide intervals that fail to capture local uncertainty variations. BayesianNN and CF-GNN ensure full coverage by expanding the intervals significantly, resulting in conservative but less informative predictions. In contrast, QpiGNN adaptively adjusts its interval widths, slightly increasing them when needed to satisfy the target coverage ($1 - \alpha = 0.9$), while preserving interval sharpness where uncertainty is low.

These trends persist on the real-world Anaheim dataset, where QpiGNN again balances calibration and informativeness. The ability to modulate interval width in response to local uncertainty allows QpiGNN to avoid both undercoverage and overconservativeness, unlike other methods that either overfit to fixed-width regimes or fail to generalize.

# L. Computational Efficiency and Complexity Analysis

Table 12 presents a comparison of computational resource usage across all models during training on real-world graph datasets, including training time, peak memory usage, and model size, all measured using an NVIDIA Tesla V100 GPU. The results highlight the trade-offs between efficiency and uncertainty estimation quality across different approaches. We present a theoretical comparison of the time complexity of all baseline models considered in this study. Let $N$ and $E$ denote the number of nodes and edges in the graph and $d$ the hidden dimension of the GNN layers.

QpiGNN, RQR$^{adj.}$-GNN, and SQR-GNN share a common two-layer GraphSAGE (Hamilton et al., 2017) backbone and differ only in their output heads or loss formulations. Their time complexity is identical $\mathcal{O}(Ed + Nd^2)$. MC dropout requires $T$ stochastic forward passes at inference time to approximate uncertainty, resulting in a total complexity of

*Table 12.* Comparison of average computational resource consumption across models during real-world dataset training.

| Model | | Education | Election | Income | Unemploy. | Twitch | Chameleon | Crocodile | Squirrel | Anaheim | Chicago |
|---|---|---|---|---|---|---|---|---|---|---|---|
| **Training Time (sec)** | SQR-GNN | 1.39 | 1.43 | 1.61 | 1.52 | 1.83 | 1.83 | 13.76 | 2.98 | 1.87 | 1.93 |
| | RQR$^{adj.}$-GNN | 2.06 | 1.99 | 1.79 | 2.39 | 2.10 | 2.00 | 31.51 | 7.81 | 1.78 | 1.71 |
| | BayesianNN | 1.32 | 1.28 | 1.31 | 1.32 | 1.34 | 1.41 | 21.34 | 5.36 | 1.48 | 1.26 |
| | MC dropout | 1.83 | 1.83 | 1.82 | 1.37 | 1.77 | 1.53 | 18.37 | 2.92 | 1.70 | 1.80 |
| | CF-GNN | 3.63 / 9.77 | 3.44 / 9.86 | 3.70 / 9.80 | 3.48 / 9.73 | 4.83 / 9.61 | - | - | - | 3.48 / 8.21 | 3.27 / 9.06 |
| | QpiGNN | 1.97 | 1.84 | 1.74 | 1.70 | 1.63 | 1.84 | 14.87 | 5.28 | 1.92 | 1.87 |
| **CPU (MB)** | SQR-GNN | 68.56 | 68.56 | 68.56 | 68.56 | 401.34 | 447.80 | 8069.80 | 2016.32 | 65.74 | 68.04 |
| | RQR$^{adj.}$-GNN | 286.70 | 286.70 | 286.70 | 286.70 | 382.78 | 425.81 | 7601.41 | 1966.33 | 82.11 | 167.08 |
| | BayesianNN | 68.47 | 68.47 | 68.47 | 68.47 | 382.77 | 425.79 | 7601.32 | 1966.30 | 65.72 | 67.99 |
| | MC dropout | 69.23 | 69.23 | 69.23 | 69.23 | 382.77 | 425.79 | 7601.61 | 1966.29 | 65.94 | 68.49 |
| | CP | 67.58 | 67.58 | 67.58 | 67.58 | 299.72 | 331.93 | 5712.37 | 1340.74 | 65.65 | 67.02 |
| | CF-GNN | 70.70 / 69.53 | 70.70 / 69.53 | 70.70 / 69.53 | 70.70 / 69.52 | 523.31 / 521.99 | - | - | - | 66.02 / 65.73 | 70.27 / 69.54 |
| | QpiGNN | 68.48 | 68.48 | 68.48 | 68.48 | 382.78 | 425.80 | 7601.44 | 1966.31 | 1306.90 | 68.00 |
| **Parameters (1,000)** | SQR-GNN | 9.28 | 9.28 | 9.28 | 9.28 | 414.27 | 409.41 | 1695.94 | 411.46 | 9.03 | 9.03 |
| | RQR$^{adj.}$-GNN | 9.22 | 9.22 | 9.22 | 9.22 | 414.21 | 409.35 | 1695.87 | 411.39 | 8.96 | 8.96 |
| | BayesianNN | 9.22 | 9.22 | 9.22 | 9.22 | 414.21 | 409.35 | 1695.87 | 411.39 | 8.96 | 8.96 |
| | MC dropout | 9.15 | 9.15 | 9.15 | 9.15 | 414.15 | 409.28 | 1695.81 | 411.33 | 8.90 | 8.90 |
| | CP | 9.15 | 9.15 | 9.15 | 9.15 | 414.15 | 409.28 | 1695.81 | 411.33 | 8.90 | 8.90 |
| | CF-GNN | 1.22 | 1.22 | 1.22 | 1.22 | 406.21 | - | - | - | 0.96 | 0.96 |
| | QpiGNN | 9.22 | 9.22 | 9.22 | 9.22 | 414.21 | 409.35 | 1695.87 | 411.39 | 8.96 | 8.96 |

*Table 13.* Generalization performance across different synthetic graph datasets. Each cell reports the average PICP and MPIW over 10 runs with a fixed width penalty $\lambda_{\text{width}} = 0.5$.

| Target | PICP | | | | | MPIW | | | | |
|---|---|---|---|---|---|---|---|---|---|---|
| | BA | ER | basic | grid | tree | BA | ER | basic | grid | tree |
| **BA** | 0.8715 | 0.9720 | 0.7769 | 0.9527 | 0.9627 | 0.5549 | 1.2381 | 1.0600 | 1.3330 | 1.1843 |
| **ER** | 0.8324 | 0.9289 | 0.6719 | 0.8627 | 0.8892 | 0.7489 | 0.6280 | 0.8749 | 0.9655 | 0.9233 |
| **basic** | 0.1000 | 0.2750 | 0.9000 | 0.8277 | 0.4313 | 0.3060 | 0.4040 | 0.4018 | 0.5048 | 0.4799 |
| **grid** | 0.7064 | 0.6385 | 0.5250 | 0.8250 | 0.6578 | 0.8479 | 0.8186 | 0.8099 | 0.8072 | 0.8335 |
| **tree** | 0.1045 | 0.5625 | 0.2495 | 0.5644 | 0.9058 | 0.3884 | 0.4299 | 0.3389 | 0.3861 | 0.3905 |

$\mathcal{O}(T \cdot (Ed + Nd^2))$. Bayesian Neural Networks (BayesianNN) introduces posterior sampling in the final layer, incurring additional cost per forward pass $\mathcal{O}(Ed + Nd^2 + Nd)$. Conformalized GNN (CF-GNN) attaches a separate multi-layer GNN module (ConfGNN) for calibration on top of the base predictor. Let $d'$ and $L$ denote the hidden dimension and number of layers in ConfGNN, respectively. The resulting complexity is $\mathcal{O}(Ed + Nd^2 + L \cdot (Ed' + Nd'^2))$. This auxiliary component significantly increases both runtime and memory requirements, which may limit scalability, particularly for large or dense graphs.

QpiGNN and its quantile-based variants (RQR, SQR) offer strong computational efficiency by avoiding repeated forward passes or additional modules. In contrast, MC dropout and CF-GNN introduce notable overhead due to ensemble-style inference and dual-network design, respectively. This trade-off underscores the practical scalability advantage of QpiGNN for efficient uncertainty estimation in graph learning.

## M. Structural Shift Generalization

To evaluate the generalization capacity of QpiGNN under distribution shift, we conduct an out-of-distribution (OOD) experiment across synthetic graph types. Specifically, we train the model on a single graph type and evaluate it on all other types, without further fine-tuning or recalibration. Table 13 presents the results. Each row indicates the graph type used for training, while columns correspond to the test graph types. We report the PICP and MPIW, averaged over 10 independent runs using a fixed-width penalty $\lambda_{\text{width}} = 0.5$. Higher PICP reflects better coverage (calibration), and lower MPIW indicates sharper, more compact intervals.

QpiGNN shows the strongest generalization when trained on expressive graphs such as *BA* and *ER*, achieving high PICP scores on multiple unseen target types (e.g., BA→ER: 0.9720, ER→grid: 0.8627). However, this often comes at the cost of

*Table 14.* Ablation results on 19 synthetic and real datasets. ✓ denotes an enabled component. Each result is averaged over 5 runs of 500 epochs. The width penalty factor is set to $\lambda_{width} = 0.5$. The first column shows PICP and the second shows MPIW. **Bold** indicates models achieving PICP $\geq 0.9$ (target coverage $1-\alpha$), and among them, the configuration with the lowest MPIW is highlighted.
*Note: The "No Loss" setting defaults to MSE loss over central prediction.*

| Model Variant | Dual | Coverage | Width | Basic | | Gaussian | | Uniform | | Outlier | | Edge | | ER | | BA | | grid | | tree | |
|---|---|---|---|---|---|---|---|---|---|---|---|---|---|---|---|---|---|---|---|---|---|
| | | | | PICP | MPIW | PICP | MPIW | PICP | MPIW | PICP | MPIW | PICP | MPIW | PICP | MPIW | PICP | MPIW | PICP | MPIW | PICP | MPIW |
| *Architecture-Level Ablation* | | | | | | | | | | | | | | | | | | | | | |
| Dual Head (learned diff) | ✓ | ✓ | ✓ | **0.91** | 0.31 | **0.92** | 0.53 | 0.88 | 0.43 | 0.88 | 0.47 | **0.93** | 0.43 | **0.98** | 0.58 | **0.98** | 0.45 | **0.98** | 0.86 | **0.96** | 0.39 |
| Fixed Margin (0.05) | ✓ | ✓ | ✓ | 0.72 | 0.10 | 0.29 | 0.10 | 0.25 | 0.10 | 0.68 | 0.10 | 0.77 | 0.10 | 0.64 | 0.10 | 0.43 | 0.10 | 0.18 | 0.10 | 0.32 | 0.10 |
| Fixed Margin (0.10) | ✓ | ✓ | ✓ | **0.94** | 0.20 | 0.51 | 0.20 | 0.51 | 0.20 | 0.68 | 0.20 | **0.97** | 0.20 | 0.84 | 0.20 | 0.73 | 0.20 | 0.38 | 0.20 | 0.67 | 0.20 |
| Single Head (low/high sep.) | ✗ | ✓ | ✓ | 0.46 | 0.07 | 0.25 | 0.09 | 0.25 | 0.10 | 0.71 | 0.23 | 0.58 | 0.07 | 0.78 | 0.19 | 0.32 | 0.11 | 0.26 | 0.14 | 0.06 | 0.02 |
| *Loss-Level Ablation* | | | | | | | | | | | | | | | | | | | | | |
| Full Loss | ✓ | ✓ | ✓ | **0.91** | 0.31 | **0.92** | 0.53 | 0.88 | 0.43 | 0.88 | 0.47 | **0.93** | 0.43 | **0.98** | 0.58 | **0.98** | 0.45 | **0.98** | 0.86 | **0.96** | 0.39 |
| Coverage Only | ✓ | ✓ | ✗ | **1.00** | 1.24 | **1.00** | 1.18 | **1.00** | 1.21 | 0.99 | 1.13 | **1.00** | 1.30 | 0.99 | 1.09 | 0.97 | 1.16 | 0.99 | 1.14 | **1.00** | 1.10 |
| Width Only | ✓ | ✗ | ✓ | 0.00 | 0.00 | 0.00 | 0.00 | 0.00 | 0.00 | 0.00 | 0.00 | 0.00 | 0.00 | 0.00 | 0.00 | 0.00 | 0.01 | 0.00 | 0.02 | 0.00 | 0.01 |
| No Loss (Sanity Check) | ✓ | ✗ | ✗ | **1.00** | 1.00 | **1.00** | 1.00 | **1.00** | 1.00 | 0.98 | 1.00 | **1.00** | 1.00 | 0.99 | 1.00 | 0.99 | 1.00 | **1.00** | 1.00 | **1.00** | 1.00 |

| Model Variant | Dual | Coverage | Width | Education | | Election | | Income | | Unemploy. | | Twitch | | Chameleon | | Crocodile | | Squirrel | | Anaheim | | Chicago | |
|---|---|---|---|---|---|---|---|---|---|---|---|---|---|---|---|---|---|---|---|---|---|---|---|
| | | | | PICP | MPIW | PICP | MPIW | PICP | MPIW | PICP | MPIW | PICP | MPIW | PICP | MPIW | PICP | MPIW | PICP | MPIW | PICP | MPIW | PICP | MPIW |
| *Architecture-Level Ablation* | | | | | | | | | | | | | | | | | | | | | | | |
| Dual Head (learned diff) | ✓ | ✓ | ✓ | **0.99** | 0.56 | **0.99** | 0.82 | **0.99** | 0.40 | **0.99** | 0.70 | 0.60 | 0.09 | 0.61 | 0.04 | **0.94** | 0.09 | 0.71 | 0.07 | **0.93** | 0.40 | **0.97** | 0.36 |
| Fixed Margin (0.05) | ✓ | ✓ | ✓ | 0.45 | 0.10 | 0.30 | 0.10 | 0.59 | 0.10 | 0.49 | 0.10 | 0.71 | 0.10 | **0.91** | 0.10 | **0.94** | 0.10 | 0.88 | 0.10 | 0.50 | 0.10 | 0.64 | 0.10 |
| Fixed Margin (0.10) | ✓ | ✓ | ✓ | 0.72 | 0.20 | 0.56 | 0.20 | 0.83 | 0.20 | 0.78 | 0.20 | **0.90** | 0.20 | **0.95** | 0.20 | **0.98** | 0.20 | **0.97** | 0.20 | 0.77 | 0.20 | 0.84 | 0.15 |
| Single Head (low/high sep.) | ✗ | ✓ | ✓ | 0.27 | 0.07 | 0.37 | 0.14 | 0.81 | 0.22 | 0.55 | 0.12 | 0.21 | 0.02 | 0.48 | 0.02 | 0.85 | 0.03 | 0.43 | 0.03 | 0.43 | 0.11 | 0.67 | 0.15 |
| *Loss-Level Ablation* | | | | | | | | | | | | | | | | | | | | | | | |
| Full Loss | ✓ | ✓ | ✓ | **0.99** | 0.56 | **0.99** | 0.82 | **0.99** | 0.40 | **0.99** | 0.70 | 0.60 | 0.09 | 0.61 | 0.04 | **0.94** | 0.09 | 0.71 | 0.07 | **0.93** | 0.40 | **0.97** | 0.36 |
| Coverage Only | ✓ | ✓ | ✗ | **1.00** | 1.12 | **1.00** | 1.13 | **1.00** | 1.11 | **1.00** | 1.12 | **1.00** | 1.17 | **1.00** | 1.01 | **1.00** | 1.03 | **1.00** | 1.06 | **1.00** | 1.13 | **1.00** | 1.11 |
| Width Only | ✓ | ✗ | ✓ | 0.00 | 0.00 | 0.00 | 0.00 | 0.00 | 0.00 | 0.00 | 0.00 | 0.00 | 0.00 | 0.21 | 0.00 | 0.29 | 0.00 | 0.08 | 0.00 | 0.08 | 0.00 | 0.05 | 0.01 |
| No Loss (Sanity Check) | ✓ | ✗ | ✗ | **1.00** | 1.00 | **1.00** | 1.00 | **1.00** | 1.00 | **1.00** | 1.00 | **1.00** | 1.00 | **1.00** | 1.00 | **1.00** | 1.00 | **1.00** | 1.00 | **1.00** | 1.00 | **1.00** | 1.00 |

wider intervals (e.g., MPIW $\geq 1.2$). In contrast, models trained on simple graphs such as *basic* or *tree* tend to under-cover other graph types despite producing narrow intervals (e.g., basic$\rightarrow$ BA: PICP = 0.1000, MPIW = 0.3060). Structurally rich graphs improve generalization but lead to wider intervals, while simple graphs yield narrower yet poorly calibrated intervals—highlighting the need for expressive source graphs in uncertainty transfer.

# N. Ablation Study Results

To better understand the contributions of individual components within QpiGNN, we conduct a comprehensive ablation study across nine synthetic graph datasets. Table 14 presents results from both architecture-level and loss-level ablations. Each configuration is evaluated over five runs for 500 training epochs, using a fixed width penalty coefficient $\lambda_{width} = 0.5$.

**Architecture-Level Ablation.** We first evaluate different architectural choices. The full *Dual Head* model with learned margin prediction (i.e., separate heads for mean and interval width) achieves strong performance across all datasets, consistently satisfying the target coverage level ($1 - \alpha = 0.9$) while minimizing interval width (lowest MPIW in most cases). In contrast:

- The Fixed Margin variants fail to adapt to dataset-specific uncertainty patterns, leading to either severe undercoverage (e.g., Gaussian, Uniform) or overestimation of width (e.g., Tree).

- The Single Head model, where upper and lower bounds are predicted independently, shows unstable behavior, often resulting in poor calibration and highly variable coverage.

These results confirm the importance of a dedicated dual-head architecture that allows flexible, learned interval widths conditioned on node features.

**Loss-Level Ablation.** Next, we assess the role of each loss component:

- The Full Loss (coverage + width + violation penalty) yields a consistent balance between calibration and compactness across all graphs.

- The Coverage-Only variant achieves perfect coverage on all datasets but produces excessively wide intervals (high MPIW and CWC), sacrificing informativeness for reliability.

- The Width-Only variant fails entirely, collapsing all intervals to zero, resulting in complete undercoverage—demonstrating that coverage loss is essential to meaningful UQ.

- The No Loss setting (MSE over center prediction, used as a sanity check) trivially learns to output wide enough intervals to contain all targets, but lacks meaningful control over sharpness or adaptive behavior.

Taken together, these results justify the design of QpiGNN's learning objective and architecture. Both the dual-head structure and the full composite loss are critical for achieving high-quality, well-calibrated, and compact prediction intervals across diverse graph topologies.

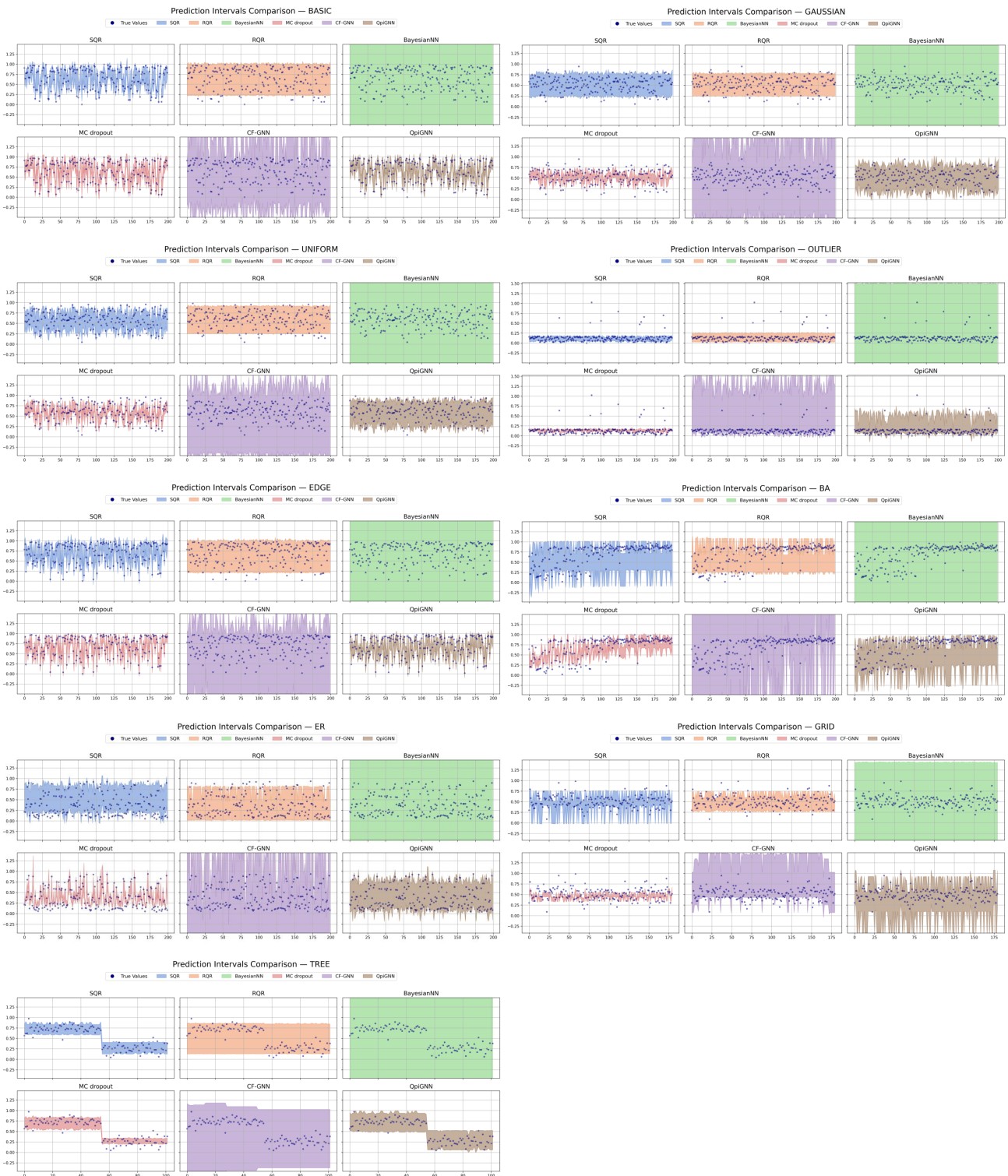

*Figure 9.* Prediction interval comparison across 9 synthetic datasets for qualitative analysis.

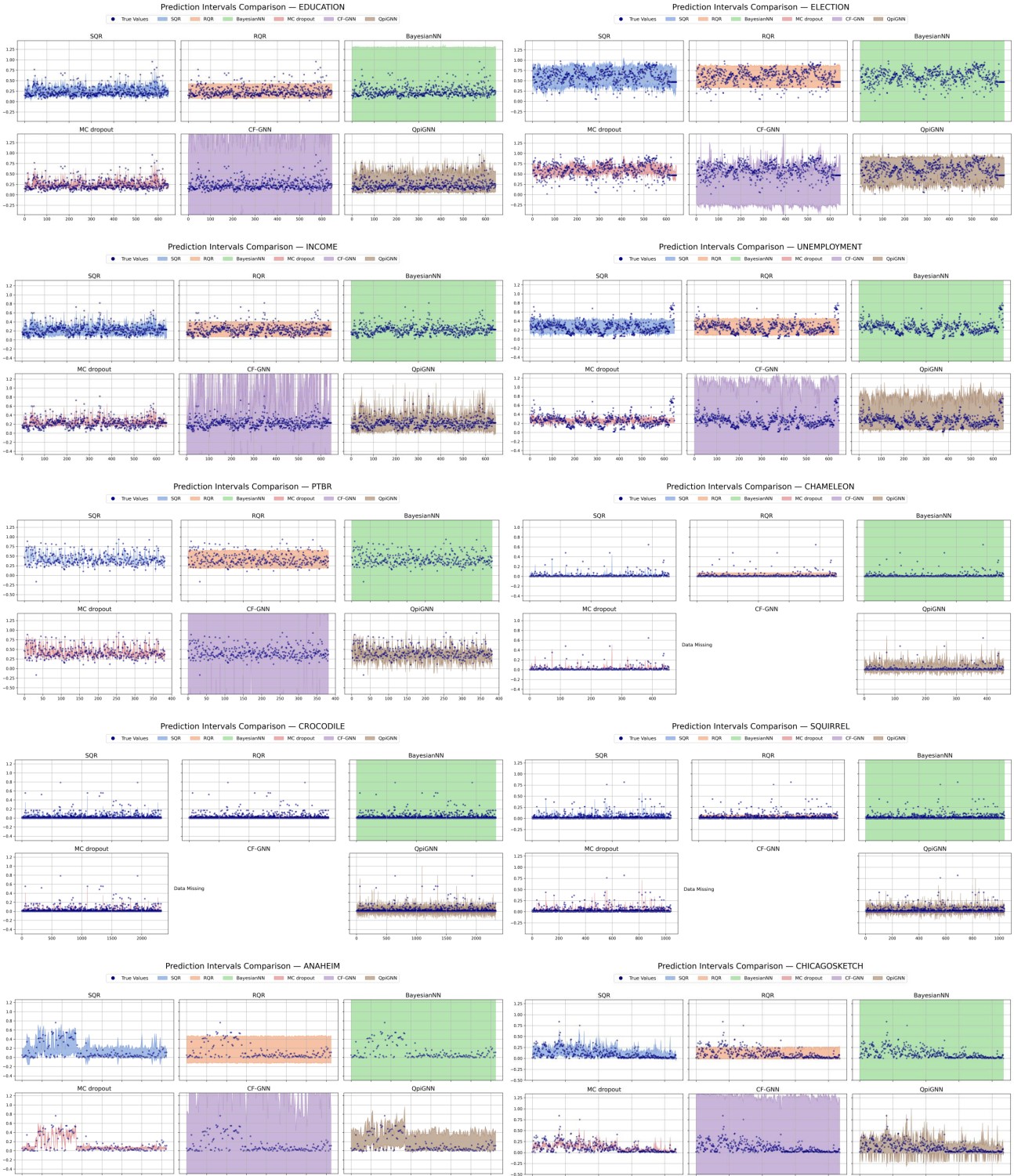

*Figure 10.* Prediction interval comparison across 10 real datasets for qualitative analysis.

