# OpenReview forum: "Quantile-Free Uncertainty Quantification in Graph Neural Networks"
_ICML.cc/2026/Conference — ICML 2026 regular_

### Official Review · Reviewer_Wsuq · 2026-02-26

**Soundness:** 4
**Presentation:** 3
**Significance:** 2
**Originality:** 3
**Overall Recommendation:** 4
**Confidence:** 3

**Summary:**

The paper proposes QpiGNN (Quantile-free Prediction Interval GNN), a framework for uncertainty quantification (UQ) in node regression tasks within Graph Neural Networks. The authors argue that existing UQ methods for graphs either rely on costly resampling (ensembles), require strong assumptions like exchangeability (Conformal Prediction), or suffer from issues like quantile crossing and oversmoothing when applied to message-passing architectures (standard Quantile Regression).

QpiGNN introduces a dual-head architecture: one head predicts the target value and the other predicts the interval width. It utilizes a "quantile-free" joint loss function that simultaneously optimizes for empirical coverage and minimizes interval width. Theoretical analysis is provided regarding asymptotic coverage and finite-sample bounds under mild assumptions. Extensive experiments on 19 datasets demonstrate that QpiGNN achieves a superior trade-off between coverage and interval width compared to baselines like SQR-GNN, CF-GNN, and Bayesian methods.

**Compliance With Llm Reviewing Policy:**

Affirmed.

**Final Justification:**

This rebuttal has addressed most of my concerns, and I recommend acceptance.

**Key Questions For Authors:**

* **1:** The paper currently lacks a detailed theoretical and empirical analysis of the model's computational complexity and memory consumption, particularly concerning large-scale graph structures. Could the authors explicitly derive how the time and space complexity scale with respect to the number of nodes ($N$) and edges ($E$)? Furthermore, empirical evidence demonstrating memory usage trends as graph size increases would significantly strengthen the claims regarding the model's efficiency.
* **2:** I have concerns regarding the stability of the method with respect to the hyperparameter $\lambda$. In datasets such as Twitch and Chameleon, the model exhibits binary behavior: when $\lambda = 0.5$, the coverage appears to fail completely (dropping well below the target), yet when $\lambda = 1.0$ (or other values), the coverage far exceeds the target. This drastic sensitivity contradicts the reliability typically expected in Uncertainty Quantification (UQ) and Conformal Prediction (CP) literature, where coverage guarantees—or at least smooth degradation—are standard. Can the authors explain the theoretical or empirical reasons for this 'all-or-nothing' behavior?
* **3:** The experimental evaluation would benefit from including more recent state-of-the-art baselines to better contextualize the contribution. For example, Residual Reweighted Conformal Prediction for Graph Neural Networks (Zhang et al, UAI).
* **4:** The experimental results currently report only mean values, omitting standard deviations. Without quantifying the variance, it is difficult to assess the stability and statistical significance of the reported improvements.

**Limitations:**

yes

**Strengths And Weaknesses:**

* **Architecture Motivation & Design:** The identification of the "entanglement" problem in single-head GNNs for UQ is insightful. The authors correctly identify that standard message passing tends to smooth out local uncertainties, leading to non-adaptive intervals. The dual-head design (separating $\hat{y}$ and $\hat{d}$) is a logical and effective architectural choice to mitigate this.
* **Comprehensive Evaluation:** The experimental setting is robust, covering 19 datasets (synthetic and real-world). The inclusion of "out-of-distribution" scenarios (noise injection, edge dropout) and non-exchangeable splits (Degree, Community) significantly strengthens the claims regarding robustness.
* **Performance:** The empirical results are strong. QpiGNN demonstrates a favorable trade-off between PICP (coverage) and MPIW (width), particularly in comparison to Conformal Prediction methods (CF-GNN) which often yield very wide intervals to satisfy coverage guarantees.
* **Theoretical Grounding:** The paper attempts to provide theoretical justification for finite-sample coverage based on concentration inequalities adapted for graphs with bounded dependencies.
 * **Weaknesses**
* **1:** The paper currently lacks a detailed theoretical and empirical analysis of the model's computational complexity and memory consumption, particularly concerning large-scale graph structures. Could the authors explicitly derive how the time and space complexity scale with respect to the number of nodes ($N$) and edges ($E$)? Furthermore, empirical evidence demonstrating memory usage trends as graph size increases would significantly strengthen the claims regarding the model's efficiency.
* **2:** I have concerns regarding the stability of the method with respect to the hyperparameter $\lambda$. In datasets such as Twitch and Chameleon, the model exhibits binary behavior: when $\lambda = 0.5$, the coverage appears to fail completely (dropping well below the target), yet when $\lambda = 1.0$ (or other values), the coverage far exceeds the target. This drastic sensitivity contradicts the reliability typically expected in Uncertainty Quantification (UQ) and Conformal Prediction (CP) literature, where coverage guarantees—or at least smooth degradation—are standard. Can the authors explain the theoretical or empirical reasons for this 'all-or-nothing' behavior?
* **3:** The experimental evaluation would benefit from including more recent state-of-the-art baselines to better contextualize the contribution. For example, Residual Reweighted Conformal Prediction for Graph Neural Networks (Zhang et al, UAI).
* **4:** The experimental results currently report only mean values, omitting standard deviations. Without quantifying the variance, it is difficult to assess the stability and statistical significance of the reported improvements.

---

> ### Author Rebuttal · Authors · 2026-03-31
>
> We thank the reviewer for the constructive feedback. We address each concern below regarding computational scalability, the sensitivity of $\lambda_{width}$ to coverage behavior, comparison with recent baselines, and the reporting of standard deviations.
>
> > **Q1. Computational Complexity and Scalability Analysis**
>
> We thank the reviewer for emphasizing scalability. QpiGNN is designed to match the efficiency of its GNN backbone, with time and space complexity of $O(Ed + Nd^2)$ and $O(Nd + E)$, respectively. The dual-head design introduces only `negligible overhead.`
>
> Importantly, QpiGNN performs inference in a `single forward pass`, whereas sampling-based methods incur an `additional $O(T)$ cost.` Empirical results confirm that QpiGNN achieves runtime and memory usage comparable to standard GNNs with minimal overhead (Appendix L, Table 12). Overall, QpiGNN maintains GNN-level efficiency while avoiding the additional cost of sampling-based approaches.
> |Model|Crocodile (Time / Memory)|Squirrel (Time / Memory)|
> |-|-|-|
> |SQR-GNN|13.76s / 8069MB| 2.98s / 2016MB|
> |RQR-GNN|31.51s / 7601MB| 7.81s / 1966MB|
> |BayesianNN|21.34s / 7601MB| 5.36s / 1966MB|
> |MC-dropout|18.37s / 7601MB| 2.92s / 1966MB|
> |QpiGNN|14.87s / 7601MB| 5.28s / 1966MB |
>
> ---
> > **Q2. Theoretical and Empirical Explanation of the 'All-or-Nothing' Behavior**
>
> We thank the reviewer for this insightful observation. The “all-or-nothing” behavior arises from the `non-convex trade-off between coverage and interval width`, controlled by $\lambda_{width}$. Small changes in $\lambda_{width}$ can shift the model between under-coverage and overly conservative intervals.
>
> Experiments show that fixed $\lambda_{width}$ leads to extreme outcomes, while annealing achieves near-target coverage with moderate width. This indicates the issue is mainly driven by `optimization dynamics rather than the objective itself.`
>
> However, this is a preliminary finding. While annealing improves stability, it `does not ensure exact calibration`; `target-aware adaptive scheduling` is a promising direction for future work.
> |Dataset|Schedule|PICP ↑|MPIW ↓|
> |-|-|-|-|
> |Chameleon|$\lambda_{width}=0.5$|0.41 ± 0.11|0.03 ± 0.01|
> ||$\lambda_{width}=0.1$|0.99 ± 0.00|0.45 ± 0.06|
> ||Linear Annealing|0.95 ± 0.01|0.20 ± 0.03|
> ||Cosine Annealing|0.96 ± 0.01|0.22 ± 0.03|
> ||Step Decay|0.96 ± 0.01|0.22 ± 0.02|
> |Twitch|$\lambda_{width}=0.5$|0.54 ± 0.08|0.09 ± 0.02|
> ||$\lambda_{width}=0.1$|0.98 ± 0.00|0.51 ± 0.03|
> ||Linear Annealing|0.93 ± 0.03|0.33 ± 0.03|
> ||Cosine Annealing|0.95 ± 0.01|0.31 ± 0.03|
> ||Step Decay|0.94 ± 0.02|0.34 ± 0.05|
>
> ---
> > **Q3. Comparison with Recent Baseline**
>
> Thank you for this suggestion. We add a contextual comparison with Zhang et al. Since public code is unavailable, `experimental conditions may differ and this should be interpreted as a qualitative reference rather than a strict benchmark.`
>
> With this caveat, QpiGNN achieves `comparable or better coverage with narrower MPIW` across all four datasets (target coverage=0.95):
> |Dataset|Method|PICP ↑|MPIW ↓|
> |-|-|-|-|
> |Anaheim|RR-GAE|0.95±0.04|1.87±0.03|
> ||Cluster-RR-GAE|0.95±0.03|1.96±0.02|
> ||QpiGNN|0.96±0.08|0.64±0.04|
> |Chicago| RR-GAE|0.95±0.04|2.27±0.03|
> ||Cluster-RR-GAE|0.95±0.04|2.23±0.03|
> ||QpiGNN|0.95±0.01|0.47±0.01|
> |Education| RR-GAE|0.95±0.03|2.19±0.03|
> ||Cluster-RR-GAE|0.96±0.04|2.06±0.03|
> ||QpiGNN|0.99±0.00|0.75±0.02|
> |Election| RR-GAE|0.95±0.03|0.90±0.03|
> ||Cluster-RR-GAE|0.96±0.02|0.92±0.03|
> ||QpiGNN|0.99±0.00|0.89±0.02|
>
> The over-coverage on Education and Election is consistent with the conservative behavior discussed in Reviewer CHSC's W2&Q1. The MPIW gap may partly reflect normalization differences; a rigorous comparison will be included in the revision.
>
> ---
> > **Q4. Reporting Standard Deviations**
>
> We agree that reporting only means is insufficient. All experiments were conducted over `5–10 random seeds`, and we now report PICP and MPIW as mean ± standard deviation.
>
> QpiGNN shows low variance with strong performance, supporting its empirical stability without claiming formal significance. The consistently `low standard deviations of QpiGNN across datasets suggest stable performance` that is unlikely to be attributable to random variation.
> |Model|Gaussian|ER|Tree |Income|Chicago|
> |-|-|-|-|-|-|
> |SQR-GNN|0.88 ± 0.03 / 0.50 ± 0.06|0.75 ± 0.06 / 0.60 ± 0.07|0.80 ± 0.03 / 0.26 ± 0.03|0.86 ± 0.04 / 0.22 ± 0.03|0.87 ± 0.07 / 0.21 ± 0.02|
> |RQR$^{adj}$-GNN|0.88 ± 0.00 / 0.53 ± 0.00|0.88 ± 0.00 / 0.77 ± 0.00|0.85 ± 0.00 / 0.68 ± 0.00|0.89 ± 0.00 / 0.36 ± 0.00|0.88 ± 0.06 / 0.30 ± 0.02|
> |ER-GNN|1.00 ± 0.00 / 1.85 ± 0.37|0.72 ± 0.10 / 1.86 ± 0.30|1.00 ± 0.00 / 11.21 ± 0.82|1.00 ± 0.00 / 3.12 ± 0.29|1.00 ± 0.00 / 3.06 ± 0.39|
> |BayesianNN|1.00 ± 0.00 / 2.98 ± 0.17|1.00 ± 0.00 / 3.01 ± 0.24|1.00 ± 0.00 / 3.00 ± 0.20|1.00 ± 0.00 / 2.97 ± 0.21|1.00 ± 0.00 / 2.99 ± 0.19|
> |QpiGNN ($\lambda=0.5$)|0.92 ± 0.01 / 0.55 ± 0.02|0.98 ± 0.08 / 0.63 ± 0.10|0.96 ± 0.01 / 0.39 ± 0.02|0.99 ± 0.00 / 0.41 ± 0.00|0.97 ± 0.02 / 0.36 ± 0.01|
>
> ---

---

> > ### Author Rebuttal · Reviewer_Wsuq · 2026-04-01
> >
> > Thank you for your answer.

---

### Official Review · Reviewer_8cTK · 2026-03-07

**Soundness:** 2
**Presentation:** 4
**Significance:** 3
**Originality:** 3
**Overall Recommendation:** 5
**Confidence:** 3

**Summary:**

This paper proposes QpiGNN, an uncertainty quantification framework for node regression with graph neural networks. The method uses a dual-head architecture that predicts both an interval center and a half-width, trained with a joint objective designed to encourage coverage while minimizing interval length. Unlike quantile-based approaches, the method does not require explicit quantile estimation or post-hoc calibration. The authors provide asymptotic coverage and width guarantees under certain assumptions and evaluate the approach on 19 synthetic and real-world graph datasets with additional ablation and robustness studies.

**Compliance With Llm Reviewing Policy:**

Affirmed.

**Final Justification:**

The authors have addressed my concerns, and I have already appropriately increased my score to an even more positive level.

**Key Questions For Authors:**

1. Can you clarify the difference between CF-GNN and CF-GNN$^{opt}$? Do the details described in the **Implementation** paragraph refer to training both the base GNN and the additional GNN used for conformalization?
2. How are train/validation/test splits defined relative to the train/validation/calibration/test splits required by CF-GNN, which relies on a separate calibration dataset?
3. There are some cases where QpiGNN severely undercovers, as seen in Table 1 with the Chameleon, Twitch, and Squirrel datasets (i.e., for more heterophilous datasets). Would using heterophilic graph learning techniques be enough to address this gap, or is there something more fundamental at play?
4. How can the theoretical assumptions (e.g., bounded difference conditions or sufficiently diverse embeddings) be assessed or validated in real-world graph datasets?

**Limitations:**

Yes

**Strengths And Weaknesses:**

**Strengths**

- The paper addresses an important problem: uncertainty quantification for node-level regression in graph neural networks, which remains relatively underexplored compared to classification settings.
- The proposed method is simple and easy to implement, relying on a dual-head architecture that jointly predicts interval centers and widths without requiring quantile regression or post-hoc calibration.
- The empirical evaluation is extensive, including 19 datasets (both synthetic and real-world), as well as ablation studies and robustness analyses across multiple graph settings.

**Weaknesses**

- The theoretical guarantees rely heavily on “mild assumptions,” including bounded and weakly dependent noise and sufficiently diverse embeddings. It is unclear how these conditions can be verified or expected to hold in real-world graph settings, particularly in graphs with strong homophily/heterophily, large communities, or hubs where node embeddings may become highly correlated.
- In the finite-sample setting (Lines 259–264), the concentration result appears to bound deviations around the **expected empirical coverage**, rather than directly around the target coverage level. This makes the guarantee weaker than methods that provide explicit finite-sample validity.
    - Suggestion: It might be nice to include some sort of table detailing the types of guarantees that can be made under what assumptions (almost like a hierarchy)
- The evaluation setup could be clarified further. Some baselines, such as CF-GNN, require a separate calibration split, which effectively reduces the amount of data available for training compared to methods that do not require calibration. It would be helpful to clarify how this difference is handled to ensure fair comparisons.

---

> ### Author Rebuttal · Authors · 2026-03-31
>
> We thank the reviewer for the careful and constructive feedback. We address each concern below regarding the verifiability of theoretical assumptions, the strength of finite-sample guarantees, the fairness of baseline comparisons, and the undercoverage on heterophilous datasets.
>
> > **W1&Q4. Verifiability of Theoretical Assumptions in Real-World Graph Settings**
>
> We thank the reviewer for highlighting this issue. We agree that assumptions such as weak dependence, bounded noise, and embedding diversity are not directly verifiable in full generality, especially in graphs with strong homophily/heterophily, hubs, or large communities.
>
> Our intention is not to claim these conditions are always checkable in practice, but rather that they provide a `tractable set of assumptions under which the theoretical analysis can be stated.` In our setting, `dependence is moderated by localized message passing` ($L=2$ GraphSAGE), leading to the graph-dependent bounded-difference term in Appendix C.5.
>
> From a practical standpoint, these assumptions can be `probed post hoc`—e.g., embedding diversity via pairwise cosine similarities, and local dependence via empirical coverage sensitivity across graph regions. We will include such diagnostics in the revision. We will also clarify that the method `remains empirically effective even when these conditions are only approximately met.`
>
> ---
> > **W2. Clarification of Finite-Sample Guarantee and Proposed Hierarchy Table**
>
> We appreciate this observation and agree with the reviewer’s interpretation. Our current finite-sample concentration result `controls deviation around the expected empirical coverage, not directly around the target coverage level itself.` This is indeed `weaker than the finite-sample validity guarantees available under stronger assumptions such as exchangeability.`
>
> We will revise the manuscript to make this distinction more explicit and avoid overstating the strength of the guarantee. We also appreciate the suggestion of a hierarchy-style summary. In the revision, we plan to add a table contrasting different guarantee regimes, for example:
> - (1) weak dependence / $\alpha$-mixing $\rightarrow$ concentration around expected empirical coverage ($O(1/\sqrt{N})$).
> - (2) i.i.d assumptions $\rightarrow$ Hoeffding-type bounds.
> - (3) exchangeability $\rightarrow$ conformal finite-sample validity.
> We believe this will help readers better understand both the scope and the limitations of our result.
>
> ---
> > **W3&Q1-2. Fair Comparison with CF-GNN and Clarification of Evaluation Setup**
>
> Thank you for requesting clarification. To ensure fairness, we kept the `overall amount of labeled data fixed across all methods.` Since CF-GNN requires a calibration split, we partitioned the original training portion at an 8:2 ratio, so that train+calibration for CF-GNN matched the training size of other methods. Validation and test sets remain identical across all methods.
>
> - (Q1) Regarding CF-GNN vs. CF-GNN$^{opt}$: CF-GNN refers to our reimplementation under the common experimental backbone and protocol. CF-GNN$^{opt}$ refers to the stronger variant reported in the original paper, which uses more extensive hyperparameter tuning. In our reimplementation, the `base GNN is first trained on the train split, after which a separate conformalization GNN is trained on the calibration split`—both under our standardized protocol.
> - (Q2) To make the split mapping explicit: the common train set is divided into train (80%) and calibration (20%) for CF-GNN, while the `validation and test sets are shared identically across all methods.` We will revise the implementation paragraph to clarify these details.
>
> ---
> > **Q3. Under-coverage on Heterophilous Datasets and Potential Remedies**
>
> We thank the reviewer for this important observation. The undercoverage on Chameleon, Twitch, and Squirrel stems from two main factors.
> - GraphSAGE's neighborhood aggregation introduces noisy signals in heterophilic graphs, leading to biased representations.
> - The globally defined coverage loss causes gradients to be dominated by well-aggregated nodes, leaving heterophilous regions with insufficient corrective signals.
>
> More fundamentally, the `global coverage objective is itself agnostic to local graph topology`—meaning structural heterophily is `not just a backbone issue but also an objective-level limitation` that cannot be resolved by simply swapping the GNN architecture. We believe two directions could address this.
> - First (`targeting the backbone issue`), heterophily-aware backbones such as H2GCN or LINKX would provide the width head with a cleaner local signal.
> - Second (`targeting the objective-level limitation`),  a node-adaptive coverage loss — weighting the violation term by a local heterophily score $h_v = 1 - \text{(label agreement with neighbors)}$—would assign higher penalty to heterophilous nodes and reduce the systematic bias.
>
> We will add this to the limitation section as a concrete future direction.
>
> ---

---

> > ### Author Rebuttal · Reviewer_8cTK · 2026-04-01
> >
> > The authors have addressed all of my concerns and questions. I will update my score from a 4 to a 5

---

### Official Review · Reviewer_yr7R · 2026-03-10

**Soundness:** 3
**Presentation:** 3
**Significance:** 2
**Originality:** 3
**Overall Recommendation:** 4
**Confidence:** 3

**Summary:**

Uncertainty quantification in node-level graph regression often relies on strong exchangeability assumptions that are violated by graph structures, or requires computationally expensive resampling and post-hoc calibration methods. QpiGNN employs a dual-head Graph Neural Network to separate the prediction of the target value from the prediction of the interval width. It optimizes a joint loss function that explicitly penalizes empirical coverage violations and interval width, removing the need for quantile-specific inputs. The authors claim QpiGNN achieves an average of 22% higher coverage and 50% narrower intervals compared to baselines across 19 synthetic and real-world datasets, demonstrating robustness to structural shifts and noise.

**Compliance With Llm Reviewing Policy:**

Affirmed.

**Final Justification:**

The rebuttal addressed all of my main concerns and changed my evaluation. Specifically, the clarification on gradient propagation through the differentiable violation surrogate resolved the ambiguity in the loss formulation, the controlled stress tests provided convincing evidence for the structural incompatibility of SQR with GNNs, and the authors' transparent acknowledgment of theoretical limitations under deep/dense graph regimes was appreciated. The paper presents a well-motivated framework with comprehensive experiments across 19 datasets and thorough ablation studies. The dual-head architecture combined with the quantile-free joint loss is a reasonable and effective design for node-level UQ. The main limitations are that the theoretical contributions are restricted to shallow/sparse settings and the problem scope (UQ for GNN regression) is somewhat narrow. Overall, I believe the paper makes a solid contribution and I have updated my score from 2 to 4.

**Key Questions For Authors:**

1. How exactly are gradients propagated through the empirical coverage term in Equation 6, given that it relies on a non-differentiable indicator function? Are you using a soft approximation, or is the gradient derived entirely from the violation penalty?

2. Why do standard quantile regression baselines (e.g., SQR) exhibit catastrophic calibration failure on basic synthetic datasets? Detail the hyperparameter tuning protocol applied to the baselines.

3. The theoretical guarantees rely on the assumption of weak dependence among nodes. Given the known issue of receptive field expansion and oversmoothing in deeper GNNs, how do these concentration bounds hold mathematically as the number of layers increases in dense graphs?

**Limitations:**

Yes.

**Strengths And Weaknesses:**

Strengths: The paper conducts an extensive empirical evaluation across 19 diverse datasets, stress-testing the model under feature noise, target noise, and structural domain shifts. The ablation study thoroughly isolates the architectural and loss-level contributions. The problem motivation is clearly articulated, and the distinction between aleatoric uncertainty components is well-mapped to the dual-head architecture. Developing robust UQ methods that bypass the strict exchangeability assumptions required by standard Conformal Prediction is a highly relevant endeavor for graph machine learning. The specific engineering assembly of a decoupled coverage and width loss applied to node-level GNN regression provides an alternative to conformal post-processing.

Weaknesses: There is a critical, unexplained flaw in the optimization of the proposed loss function; Equation (6) heavily relies on an empirical coverage term defined using non-differentiable indicator functions, and the authors fail to explain how backpropagation handles this step function. The baseline comparisons appear heavily skewed, with established methods like SQR yielding catastrophic calibration failures on basic datasets, strongly indicating a strawman baseline setup.

---

> ### Author Rebuttal · Authors · 2026-03-31
>
> We thank the reviewer for the detailed feedback. We address each concern below regarding the gradient propagation through the coverage term, the validity of baseline comparisons, and the theoretical guarantees under increasing GNN depth.
>
> > **W1&Q1. Gradient Propagation via Differentiable Surrogate for the Coverage Term**
>
> We thank the reviewer for pointing this out. We agree that Eq. (6), as written, could be misread as the loss directly optimized during training, and we will clarify this more explicitly in the revision.
>
> Our training `does not backpropagate through the empirical coverage term` $\hat c$, since it contains a non-differentiable indicator function. Instead, gradients are propagated through a `differentiable surrogate based on the interval violation penalty`:
> $$\hat \ell_{viol} = \frac{1}{|\mathcal{V}|}\sum_{v \in \mathcal{V}}\Bigl(\max(0, \ \hat y^{low}_v - y_v) + \max(0, \ y_v - \hat y^{up}_v) \Bigr)$$
>
> This term provides nonzero gradients whenever the target falls outside the predicted interval, with `magnitude proportional to the violation distance.` The empirical coverage discrepancy term $(\hat{c} - (1-\alpha))^2$ is used as a `statistical objective/monitoring quantity rather than as a differentiable backpropagation target.` We will revise Section 4.3 and Algorithm 1 to make this distinction explicit and avoid ambiguity.
>
> ---
> > **W2&Q2. Fair Baseline Evaluation and Technical Cause of SQR Failure**
>
> We appreciate the reviewer's concern. All methods share the same backbone, optimizer, and data splits, with identical hyperparameter configuration (learning rate 0.001, hidden dimensions 64, layers 2; full details in Appendix I). The observed "catastrophic failure" is therefore not a tuning artifact, but stems from `fundamental structural incompatibilities` between standard QR losses and message passing. SQR was originally designed for i.i.d. MLP settings, where $\tau$-conditioning is applied to a fixed input representation.
>
> When adapted to GNNs, SQR faces two key structural barriers:
> - `Single-head Coupling`: Message passing entangles prediction and uncertainty signals, washing out node-wise heteroskedasticity crucial for valid intervals — especially in heterophilic graphs.
> - `Node-wise Loss vs. Graph Dependence`: Pinball loss assumes i.i.d. samples, but GNN representations are `structurally dependent through the adjacency matrix`, causing unstable gradients and convergence to trivial solutions.
>
> Stress tests under identical settings confirm this (Table 2): SQR-GNN's coverage drops as low as `0.49—well below the nominal target of 0.90`—while QpiGNN maintains stable coverage across all perturbation types, confirming that the failure is `structural rather than a tuning artifact.`
> |Perturbation Type|QpiGNN (Ours)|SQR-GNN|RQR-GNN|
> |-|-|-|-|
> |Feature Noise|0.89–0.92 / 0.66–0.83|0.65–0.76 / 0.69–0.81|0.84–0.85 / 0.76–0.78|
> |Target Noise|0.96–1.00 / 0.44–1.05|0.49–0.98 / 0.52–0.99|0.95–0.98 / 0.87–1.32|
> |Edge Dropout|0.84–0.91 / 0.21–0.23|0.53–0.82 / 0.44–0.50|0.86–0.89 / 0.80–0.84|
>
> This supports our claim that adapting quantile regression to graphs requires more than simply applying MLP-based losses to GNN backbones. We will add an explicit discussion of these structural barriers in the revision.
>
> ---
> > **Q3. Validity of Concentration Bounds Under GNN Depth and Graph Density**
>
> Thank you for this question. Our concentration results in Proposition 4.1 rely on weak dependence and bounded-difference assumptions, which are most plausible in the regime considered in our experiments: a 2-layer GraphSAGE model on sparse-to-moderate density graphs. With $L=2$, each node depends only on a limited local neighborhood, helping constrain the effective dependence structure.
>
> Mathematically, this dependence is captured through a graph-dependent bounded-difference term $\delta_G$ in Appendix C.5, which yields a concentration rate of $O(1/\sqrt{N})$ under the stated assumptions. As depth $L$ increases, however, the receptive field expands and $\delta_G$ `grows accordingly—weakening the concentration rate` and `potentially rendering the` $O(1/\sqrt{N})$ `bound vacuous` in deep or dense graph settings. In the extreme case of oversmoothing, where $\mathbf h_v^{(L)} \approx \phi(\bar{\mathbf x}_{\mathcal N_L(v)})$ becomes nearly identical across nodes, the `bounded-difference assumption effectively breaks down`, as individual node perturbations propagate across the entire graph.
>
> We acknowledge this as an important limitation of the current theory. The present framework is `intended for localized message passing settings`, and extending concentration guarantees to deeper or denser GNN regimes — for instance, by incorporating `mixing time arguments or graph spectral conditions` to bound $\delta_G$ more tightly—is a meaningful direction for future work. We will revise the discussion in the paper to make this scope and limitation explicit.
>
> ---

---

> > ### Author Rebuttal · Reviewer_yr7R · 2026-04-02
> >
> > The authors have addressed all of my concerns. I will update my score from 2 to 4.

---

### Official Review · Reviewer_CHSC · 2026-03-12

**Soundness:** 3
**Presentation:** 3
**Significance:** 3
**Originality:** 2
**Overall Recommendation:** 4
**Confidence:** 3

**Summary:**

This paper studies node-level uncertainty quantification for graph neural networks through calibrated prediction intervals. It proposes QpiGNN, a “quantile-free” framework that predicts a center and an interval half-width with a dual-head GNN, and trains them jointly using a coverage-width objective. This design avoids quantile-level inputs, resampling, and post-hoc calibration, while producing calibrated and compact intervals end-to-end. The paper also provides theoretical discussion on coverage and near-optimal interval width, and experiments shows better coverage and efficiency tradeoff.

**Compliance With Llm Reviewing Policy:**

Affirmed.

**Key Questions For Authors:**

- The reported results suggest that the method is often conservative: although the nominal coverage is set to 0.90, the achieved PICP is frequently around 0.97–1.00 on several datasets. Do the authors believe this is mainly due to the simplicity of some datasets, or does it reflect a systematic conservative bias of the proposed objective?
- In Figure 5 and the appendix, while the method remains fully covered, there appears to be a sharp drop in coverage between 0.5 and 0.6. At the same time, the MPIW also decreases accordingly. Does this indicate that the current hyperparameter search may be too coarse? Would a more careful or adaptive parameter search strategy be necessary to better characterize the coverage–width trade-off?

**Limitations:**

yes

**Strengths And Weaknesses:**

Strengths:

- The framework is simple and easy to understand: a shared GNN encoder, one head for prediction, one head for interval width, and a joint loss over empirical coverage and interval width. The writing is also clear and easy to follow.
- The method is data-efficient, avoiding an extra split of calibration data.
- The experiments are comprehensive and supportive.

Weaknesses
- The main methodological novelty seems to lie in the alternative optimization objective. By contrast, the dual-head architecture appears fairly standard and does not seem specific to graphs or GNNs. In my view, the graph-specific aspects mainly arise in the theoretical motivation and analysis, rather than in the architectural design itself.
- From a UQ perspective, over-coverage appears quite prevalent in the experimental results. Although this is somewhat acceptable relative to the reported baselines, it is still not an ideal outcome, especially when considering the trade-off with interval efficiency. High coverage that comes from overly wide intervals is less convincing than coverage that is better aligned with the nominal target.

---

> ### Author Rebuttal · Authors · 2026-03-31
>
> We thank the reviewer for the valuable feedback. We address each concern below regarding the graph-specificity of our architecture, the conservative coverage behavior, and the coverage–width trade-off curve.
>
> > **W1. Graph-Specific Necessity of the Dual-Head Architecture**
>
> We thank the reviewer for this point. We agree that the dual-head structure is not novel in form; our claim is that `its necessity is graph-specific`, arising from a conflict induced by message passing that `does not exist in MLP settings.`
>
> In a single-head GNN, both $\hat y_v = f_{pred}(\mathbf h_v^{(L)})$ and $\hat d_v = f_{width}(\mathbf h_v^{(L)})$ are derived from the same aggregated representation, so the total gradient $\partial\mathcal L/\partial\mathbf h_v^{(L)} = \partial\mathcal L_{pred}/\partial\mathbf h_v^{(L)} + \partial\mathcal L_{width}/\partial\mathbf h_v^{(L)}$ forces a single representation to simultaneously encode neighborhood-smoothed features (needed for $\hat y_v$) and node-local noise structure $\text{Var}(\epsilon_v)$ (needed for $\hat d_v$). As aggregation depth increases, $\mathbf h_v^{(L)} \approx \phi(\bar{\mathbf x}_{\mathcal N_L(v)})$, so the local uncertainty signal is progressively diluted — a phenomenon that does not occur in MLP settings where $\mathbf h_i = \phi(\mathbf x_i)$ preserves each sample's local structure independently.
>
> The dual-head design separates these gradient paths by routing width estimation through a skip connection: $\hat d_v = f_{width}(\mathbf h_v^{(L)},\ \mathbf h_v^{(0)})$, where $\mathbf h_v^{(0)} = \mathbf x_v$. This allows $\partial\mathcal L_{width}/\partial\mathbf h_v^{(0)}$ to access the node-local signal directly, while $\partial\mathcal L_{pred}/\partial\mathbf h_v^{(L)}$ continues to benefit from neighborhood aggregation — `eliminating the conflict structurally.`
> The ablation in Section 4.2 confirms that this separation `consistently improves the PICP–MPIW trade-off across all datasets.`
>
> ---
> > **W2&Q1. On Conservative Coverage and the Coverage–Width Trade-off**
>
> We appreciate this concern. The observed over-coverage is primarily driven by two factors: dataset noise level and the sensitivity of fixed λ_width to optimization dynamics, `rather than a systematic bias of the objective itself.` In short, neither factor reflects a systematic bias of the objective; rather, they are dataset-dependent and optimization-dependent phenomena that can be mitigated through adaptive $\lambda$ scheduling.
>
> - First, on noisier real-world datasets, higher intrinsic uncertainty causes the model to produce wider intervals to satisfy coverage, naturally pushing PICP above the nominal target.
> - Second, and more critically, fixed $\lambda_{width}$ tends to produce extreme outcomes: larger values (e.g., $\lambda_{width}=0.5$) risk under-coverage, while smaller values (e.g., $\lambda_{width}=0.1$) lead to overly conservative intervals — an `"all-or-nothing"` behavior stemming from the non-convex interaction between the coverage and width terms (see also our response to Reviewer Wsuq, Q2, which includes supporting experiments on Chameleon and Twitch datasets). This means that a conservatively chosen fixed λ_width, while safe, systematically inflates coverage.
>
> Importantly, Table 1 shows that QpiGNN achieves higher or comparable coverage with substantially narrower MPIW than baselines such as BayesianNN and CF-GNN, confirming that over-coverage is `not simply due to indiscriminate interval inflation.`
>
> Moreover, as detailed in our response to Reviewer Wsuq Q2, preliminary experiments with $\lambda$-annealing strategies (linear, cosine, step decay) achieve near-target PICP (~0.93–0.96) with moderate interval width, suggesting that the conservativeness is `addressable through adaptive scheduling rather than being inherent to the objective.` We will clarify this distinction in the revision.
>
> ---
> > **Q2. On the Phase Transition in Coverage–Width Trade-off**
>
> Thank you for raising this point. The sharp change around $\lambda_{width} \approx 0.5$ may indeed suggest that the coverage–width trade-off is sensitive in that region. Our current interpretation is that this behavior is related to the non-convex joint loss, where the relative influence of the violation and width terms can shift rapidly near certain thresholds.
>
> At the same time, we agree that a finer characterization of this transition would be valuable. Our current search was `not based on a very coarse manual grid`; we used `Bayesian optimization` to identify dataset-specific $\lambda^{opt}$ values (Appendix E, Table 6, Fig. 7). These results suggest that the preferred $\lambda_{width}$ `varies across datasets`, with synthetic datasets often favoring values around 0.4–0.5 and noisier real-world datasets favoring smaller values.
>
> We will clarify this point in the revision; for experiments with `adaptive` $\lambda$ `scheduling` that address this sensitivity, we refer to our response to Reviewer Wsuq Q2.
>
> ---

---

> > ### Author Rebuttal · Reviewer_CHSC · 2026-04-05
> >
> > The authors have addressed all of my concerns and questions.

---

### Decision · Program_Chairs · 2026-04-30

**Decision:**

Accept (regular)

**Comment:**

This paper proposes QpiGNN, a framework for uncertainty quantification in node-level regression for Graph Neural Networks. The method introduces a dual-head architecture that jointly predicts target values and interval widths, optimized via a quantile-free loss designed to balance empirical coverage and interval tightness. The work is well-motivated, addressing known limitations of existing approaches such as conformal methods, quantile regression, and ensemble-based techniques.

All reviewers provide consistently positive evaluations, and importantly, all concerns have been satisfactorily addressed following the rebuttal.

Reviewers highlight:

-   A clean and effective design, with the dual-head architecture and joint loss offering a principled alternative to existing UQ methods.
-   Strong empirical validation, including experiments on 19 datasets and comprehensive ablation studies.

Some limitations remain, such as the theoretical analysis is restricted to relatively simplified settings (e.g., shallow or sparse regimes). However, these are viewed as non-blocking limitations, and the paper’s empirical strength and methodological clarity outweigh these concerns.